# Robust detection of translocations in lymphoma FFPE samples using targeted locus capture-based sequencing

Amin Allahyar [1], Mark Pieterse[1,9], Joost Swennenhuis[2,9], G. Tjitske Los-de Vries [3,9], Mehmet Yilmaz[2], Roos Leguit[4], Ruud W. J. Meijers[4], Robert van der Geize[5], Joost Vermaat[6], Arjen Cleven[7], Tom van Wezel [7], Arjan Diepstra [8], Léon C. van Kempen [8], Nathalie J. Hijmering[3], Phylicia Stathi[3], Milan Sharma[1], Adrien S. J. Melquiond [1], Paula J. P. de Vree[1], Marjon J. A. M. Verstegen[1], Peter H. L. Krijger [1], Karima Hajo[2], Marieke Simonis[2], Agata Rakszewska[2], Max van Min[2], Daphne de Jong[3], Bauke Ylstra [3], Harma Feitsma[2], Erik Splinter [2✉] & Wouter de Laat [1✉]

In routine diagnostic pathology, cancer biopsies are preserved by formalin-fixed, paraffin-embedding (FFPE) procedures for examination of (intra-) cellular morphology. Such procedures inadvertently induce DNA fragmentation, which compromises sequencing-based analyses of chromosomal rearrangements. Yet, rearrangements drive many types of hematolymphoid malignancies and solid tumors, and their manifestation is instructive for diagnosis, prognosis, and treatment. Here, we present FFPE-targeted locus capture (FFPE-TLC) for targeted sequencing of proximity-ligation products formed in FFPE tissue blocks, and PLIER, a computational framework that allows automated identification and characterization of rearrangements involving selected, clinically relevant, loci. FFPE-TLC, blindly applied to 149 lymphoma and control FFPE samples, identifies the known and previously uncharacterized rearrangement partners. It outperforms fluorescence in situ hybridization (FISH) in sensitivity and specificity, and shows clear advantages over standard capture-NGS methods, finding rearrangements involving repetitive sequences which they typically miss. FFPE-TLC is therefore a powerful clinical diagnostics tool for accurate targeted rearrangement detection in FFPE specimens.

[1] Oncode Institute & Hubrecht Institute-KNAW and University Medical Center Utrecht, Utrecht, the Netherlands. [2] Cergentis BV, Utrecht, the Netherlands. [3] Amsterdam UMC-Vrije Universiteit Amsterdam, Department of Pathology and Cancer Center Amsterdam, Amsterdam, the Netherlands. [4] University Medical Centre Utrecht, Department of Pathology, Utrecht, the Netherlands. [5] Laboratorium Pathologie Oost-Nederland, Hengelo, the Netherlands. [6] Leiden University Medical Centre, Department of Hematology, Leiden, the Netherlands. [7] Leiden University Medical Center, Department of Pathology, Leiden, the Netherlands. [8] University of Groningen, University Medical Centre Groningen, Department of Pathology & Medical Biology, Groningen, the Netherlands. [9] These authors contributed equally: Mark Pieterse, Joost Swennenhuis, G. Tjitske Los-de Vries. ✉email: erik.splinter@cergentis.com; w.laat@hubrecht.eu

Structural variation (SV) in the genome is a recurring hallmark of cancer[1,2]. Translocations (genomic rearrangements between chromosomes) in particular are found as recurrent drivers in many types of hematolymphoid malignancies. They are also increasingly appreciated in various types of solid tumors, such as lung- and prostate cancer and soft tissue sarcomas, serving as diagnostic, prognostic, and even predictive parameters to guide treatment choice. Translocation analysis of specific sets of target genes is therefore increasingly implemented in routine diagnostic workflows for these malignancies. Diagnostic pathology practice is highly dependent on formalin-fixation and paraffin embedding (FFPE) procedures[3]. The resulting FFPE specimen blocks provide a long-term preservation method and are particularly suitable for morphological assessment, including immunohistochemistry and in situ hybridization techniques (ISH). Currently, fluorescence in situ hybridization (FISH) is the "gold standard" for translocation detection in lymphoma FFPE samples. Although this method is commonly applied worldwide and successful in many instances, it has various limitations. FISH assessment is reliant on sufficient morphology. Therefore, crushing artifacts, poor fixation, extensive necrosis, and apoptosis, that frequently impair morphology, often preclude reliable interpretation. Furthermore, even though FISH assays can be routinely performed in an automated fashion identical to immunohistochemistry, the analysis of the results and rearrangement detection is largely performed manually, which is labor intensive, error prone, and expensive. Moreover, FISH assessment may be difficult, equivocal, or subjective in case of uncommon breakpoints, polysomies, or deletions that result in complex patterns of fluorescent signals[4,5]. The routinely used break-apart FISH method fails to identify translocation partners, whereas fusion-FISH is only applicable in specific situations where the translocation partner is known, such as the *MYC-IGH* translocation. Knowing the exact composition of the rearrangement is imperative information that often delineates tumor progression behavior and its subclassification[6]. Finally, FISH analyses cannot be multiplexed.

More recently, next-generation sequencing (NGS) DNA capture methods have been introduced for rearrangement detection in selected gene panels in FFPE samples, which makes it possible to detect breakpoints at base-pair resolution and identify translocation partner genes[7–10]. However, such methods rely on capturing unambiguous fusion-reads, which can be challenging when non-unique sequences flank the breakpoint[11]. This is a common situation, especially for translocations in malignant lymphoma that typically involve immunoglobulin and T-cell receptor genes as translocation partners to oncogenes[12]. RNA-based detection methods are another approach for rearrangement detection in FFPE material and currently introduced in daily clinical practice for those rearrangements that result in a chimeric or altered RNA product, as is typical for soft tissue tumors[13–15]. RNA is less stable than DNA, which sometimes could affect the performance of RNA-based diagnostic methods in FFPE specimens[16]. Furthermore, RNA-based detection methods cannot detect rearrangements in non-coding sequences that drive cancer through regulatory displacement effects. This is most often the case in malignant lymphoma, in which immunoglobulin- and T-cell receptor enhancer sequences mediate overexpression of further unaltered oncogenes. Taken together, there is still a clear need in daily diagnostic pathology practice for methodologies that more robustly detect and precisely characterize translocations in FFPE specimens.

Importantly, the formalin fixation and (unscheduled) DNA fragmentation in pathological tissue processing are obligatory steps in proximity-ligation (or "chromosome conformation capture") methods. Originally invented to study chromosome folding[17], proximity-ligation methods use formalin-mediated fixation followed by in situ DNA fragmentation and ligation, to fuse DNA fragments that are most proximal within the cell nucleus. Then NGS and quantitative analyses of ligation products can provide a relative estimate for contact frequencies between pairs of sequences in the cell population and thereby enable the analysis of recurrent chromosome folding patterns. The most dominant factor that determines the contact frequency between a pair of DNA sequences is their linear adjacency on the same chromosome, whereby such contact frequency decays exponentially with increased linear separation between the two DNA sequences. Intriguingly, genomic rearrangements change the linear sequence of chromosomes and thereby alter DNA contact patterns that are generated in proximity-ligation methods. Based on this understanding, variants of proximity-ligation methods have been introduced as powerful technologies for the identification of genomic rearrangements[18–23]. Proof-of-concept that proximity-ligation methods can also detect SVs in FFPE material was recently provided in a non-blind study that applied a Hi-C protocol (i.e., a genome-wide variant of proximity-ligation assays) to 15 FFPE tumor samples. In most cases, this method (called "Fix-C") gave visually appreciable altered contact frequencies in genes previously scored to harbor rearrangement by FISH[24]. While potentially relevant to identify previously uncharacterized rearranged genes, such a genome-wide analysis requires expensive deep sequencing that is less relevant to clinical settings where the identification of rearrangements in selected genes with known clinical significance is required.

Here, we present FFPE-targeted locus capture (FFPE-TLC), which uses in situ ligation of crosslinked DNA fragments, combined with oligonucleotide probe sets to selectively pull down, sequence, and analyze the proximity-ligation products of genes with known clinical significance. FFPE-TLC was blindly applied to 149 lymphoma and control FFPE samples, obtained by resections or needle biopsies. Rearrangements were automatically scored using "PLIER" (Proximity-Ligation based IdEntification of Rearrangements), a dedicated computational and statistical framework that processes FFPE-TLC sequenced datasets and identifies rearrangement partners of target genes based on their significantly enriched proximity-ligation products (see Methods). Comparison of FISH and FFPE-TLC results show that FFPE-TLC outperforms FISH in specificity, sensitivity, and sequence details provided on the detected rearrangements. As compared to capture-NGS, FFPE-TLC offers the clear advantage of detecting rearrangements having non-unique sequences flanking the breakpoint, which are missed by capture-NGS. Therefore, FFPE-TLC is a powerful tool for SV detection in FFPE samples in malignant lymphoma and other translocation-mediated malignancies.

## Results

**Study design and sample preparation for FFPE-TLC.** A detailed, step-by-step protocol for FFPE-TLC is provided in Supplementary information. In brief, for FFPE-TLC a 2–10 µm FFPE scroll of a representative tumor sample is deparaffinized and mildly de-crosslinked to enable in situ DNA digestion by a restriction enzyme (NlaIII) that creates fragments with a median size of 141 bp. After in situ ligation and overnight reverse crosslinking, on day two standard protocols for (probe-based) hybridization capturing are followed (see also Methods for details) and resulting libraries are sequenced in an Illumina sequencing machine (Fig. 1A and Suppl. Fig. 1). In our current probe panel for lymphoma, we targeted the *BCL2*, *BCL6,* and *MYC* genes, bus also included the immunoglobulin loci *IGH, IGK, IGL,* and other loci implicated in hematolymphoid

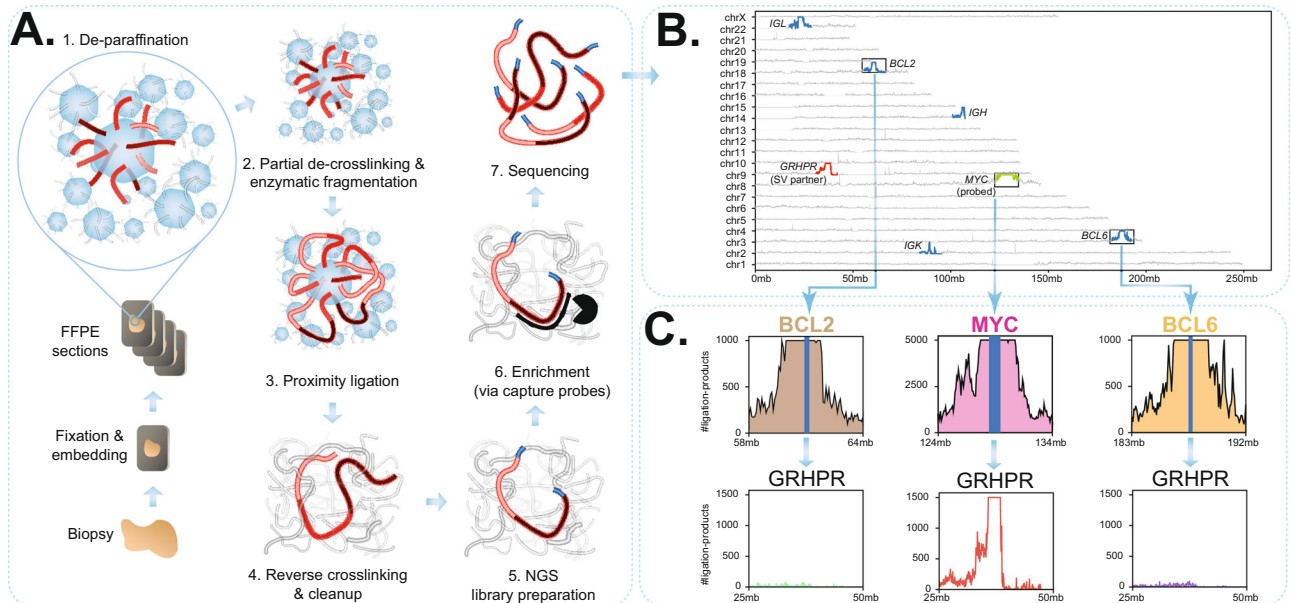

**Fig. 1 Overview of FFPE-TLC and an example of the identified rearrangements. A** Schematic overview of the FFPE-TLC workflow. (1) Through sample fixation, spatially proximal sequences (red) are preferentially cross-linked. Next, paraffin is removed and the sample section is permeabilized to allow enzymes to access the DNA. (2) The DNA is fragmented using NlaIII and then (3) ligated, which results in concatenates of co-localizing DNA fragments. (4) After crosslink reversal and DNA purification, (5) the DNA is subjected to next-generation sequencing library preparation. (6) Sequences of interest are enriched using hybrid capture probes. (7) The prepared library is paired-end Illumina sequenced. **B** Genome-wide coverage of fragments retrieved from a typical FFPE-TLC experiment targeting MYC, BCL2, and BCL6. Shown in blue is the coverage seen at the (±5 Mb) genomic intervals targeted by the capture probes. The rearranged region to the MYC gene (in green) is identified by the concentration of fragments clustered around the GRHPR gene (chr9:31mb–42mb), shown in red. **C** The probe sets used in FFPE-TLC not only retrieve the probe-complementary genomic sequences (in blue), but also megabases of its flanking sequences (i.e., the proximity-ligation products), shown for MYC (pink), BCL2 (brown), and BCL6 (orange). In case of a rearrangement (MYC-GRHPR in this case), the corresponding capture probes also retrieve fragments originating from the rearrangement partner (GRHPR, in red). This is not the case for regions that do not harbor any rearrangement (e.g., BCL2 in brown or BCL6 in orange), as shown for the GRHPR locus.

malignancies (Supplementary Data 1). For sequencing, per gene of interest we aim for one million on-target reads, which allows robust detection of rearrangements even if present in only 5% of the cells (see below). After sequencing and read mapping, a dedicated algorithm called PLIER, introduced below, searches per target locus for genomic intervals with significantly increased coverage of proximity ligation products, being their candidate rearrangement partners. To unequivocally decide whether this locus is directly fused to the target locus of interest, the corresponding contact matrix between the target locus and PLIER-identified candidate partner is inspected. The entire FFPE-TLC procedure, from FFPE scroll to diagnosis, currently takes 7 days (1 day sample processing for proximity ligation, 2 days library preparation and probe pulldown, 1 day sequencing and 3 days for read mapping, data analyses, and generation of final reports). With further automation and streamlined procedures, we expect that the entire procedure can be performed within 4–6 days.

We applied FFPE-TLC to 129 lymphoma tumor samples selected for the presence or absence of rearrangements involving *MYC*, *BCL2*, or *BCL6*, as originally detected by FISH (Table 1). Additionally, 20 FFPE samples from reactive lymph nodes (mostly from breast cancer patients) were included that were not analyzed by FISH but were expected to be devoid of rearrangements in the six target genes. Samples were provided by five different medical centers in the Netherlands and differed in tissue block age (Supplementary Data 2). All 149 samples were anonymized and therefore, the presence or absence of rearrangements in any of the target genes were hidden from us in this (blind) study. To illustrate results, Fig. 1B shows a genome-wide coverage of sequences retrieved from a typical FFPE-TLC experiment. A closer inspection of sequences captured at and

flanking the probe-targeted loci of *MYC*, *BCL2*, or *BCL6* (Fig. 1C) highlights the added value of combining NGS capture with proximity-ligation for rearrangement detection: not only are the probe-complementary genomic sequences (in blue) retrieved efficiently by FFPE-TLC, it also strongly enriches megabases of the flanking sequences (i.e., the proximity-ligation products, shown in Fig. 1C for *MYC* (pink), *BCL2* (brown), and *BCL6* (orange)). Since rearrangements with target loci juxtapose them to different flanking sequences, rearranged partner loci show an increased density of proximity-ligation sequences in FFPE-TLC and therefore can be uncovered. This phenomenon is depicted in Fig. 1B where *MYC* (in green) forms an unusually large number of proximity-ligation products with a locus containing the *GRHPR* gene (in red), indicative of tumor cells carrying this translocation[25].

**Automated rearrangement detection based on proximity ligation datasets**. To objectively identify rearrangement partner genes in FFPE-TLC datasets in an automated fashion we developed a computational pipeline called PLIER (**P**roximity-**L**igation based **IdE**ntification of **R**earrangements). A detailed description of the concepts, variables, and considerations behind PLIER is provided in the Methods section and graphically explained in Suppl. Fig. 2. In brief, PLIER initially demultiplexes sequenced FFPE-TLC samples into multiple FFPE-TLC datasets where each dataset consists of proximity-ligation products that are captured by a specific targeted gene (e.g., MYC). Then, for a given FFPE-TLC dataset (of a target gene), PLIER evaluates the density of proximity-ligation products across the genome to assign and compare an observed and expected proximity score to genomic

**Table 1 Comparison between FISH diagnoses and FFPE-TLC results.**

| MYC | | FFPE-TLC | | | | |
|---|---|---|---|---|---|---|
| | | MYC-IGH | MYC-IGL | MYC-IGK | MYC-others | MYC negative |
| **Control** | Negative (n = 20) | 0 | 0 | 0 | 0 | 20 |
| | **BCL2** | BCL2-IGH | BCL2-IGL | BCL2-IGK | BCL2-others | BCL2 negative |
| | Negative (n = 20) | 0 | 0 | 0 | 0 | 20 |
| | **BCL6** | BCL6-IGH | BCL6-IGL | BCL6-IGK | BCL6-others | BCL6 negative |
| | Negative (n = 20) | 0 | 0 | 0 | 0 | 20 |
| MYC | | FFPE-TLC | | | | |
| | | MYC-IGH | MYC-IGL | MYC-IGK | MYC-others | MYC negative |
| FISH | Positive (n = 49) | 30 | 4 | 1 | 12 | 2 |
| | Negative (n = 75) | 0 | 0 | 0 | 2 | 73 |
| | Inconclusive (n = 1) | 0 | 0 | 0 | 0 | 1 |
| | No data (n = 24) | 0 | 0 | 0 | 0 | 24 |
| BCL2 | | FFPE-TLC | | | | |
| | | BCL2-IGH | BCL2-IGL | BCL2-IGK | BCL2-others | BCL2 negative |
| FISH | Positive (n = 31) | 30 | 0 | 1 | 0 | 0 |
| | Negative (n = 63) | 0 | 0 | 0 | 0 | 63 |
| | Inconclusive (n = 3) | 0 | 0 | 0 | 0 | 3 |
| | No data (n = 52) | 3 | 0 | 0 | 0 | 49 |
| BCL6 | | FFPE-TLC | | | | |
| | | BCL6-IGH | BCL6-IGL | BCL6-IGK | BCL6-others | BCL6 negative |
| FISH | Positive (n = 29) | 12 | 3 | 0 | 14 | 0 |
| | Negative (n = 61) | 2 | 0 | 0 | 1 | 58 |
| | Inconclusive (n = 3) | 1 | 0 | 0 | 1 | 1 |
| | No data (n = 56) | 2 | 2 | 0 | 2 | 50 |

Quantitative overview of samples with FISH diagnosis horizontally and FFPE-TLC calls (using PLIER) vertically. Note that 'inconclusive' FISH results refer to samples carrying an unusual or uneven number of FISH signals.

intervals and calculate an enrichment score. For this, PLIER initially splits the reference genome into equally spaced genomic intervals (e.g., 5 kb or 75 kb bins) and then calculates for every interval a "proximity frequency" that is defined by the number of segments within that genomic interval that are covered by at least one fragment (i.e., a proximity-ligation product)." Proximity scores" are then calculated by Gaussian smoothing of proximity frequencies across each chromosome to remove very local and abrupt increase (or decrease) in proximity frequencies that are most likely spurious. Next, an expected (or average) proximity score and a corresponding standard deviation are estimated for genomic intervals with similar properties (e.g., genomic intervals present on trans chromosomes) by in silico shuffling of observed proximity frequencies across the genome followed by a Gaussian smoothing across each chromosome. Finally, a z-score is calculated for every genomic interval using its observed proximity score and the related expected and standard deviation of proximity scores. By combining z-scores calculated from multiple scales (i.e., interval widths such as 5 kb and 75 kb), a scale-invariant *enrichment score* is calculated (see Methods for more details). This scale-invariant enrichment score is used to recognize genomic intervals with elevated clustering of observed ligation products, being prime candidate rearrangement partners of the targeted gene. We initially identified the optimal parameters for PLIER through a comprehensive optimization procedure (see Methods for details on the optimization procedure). We then applied PLIER to all 149 samples to search for rearrangements involving the three clinically relevant targeted genes *MYC, BCL2,* and *BCL6*. An overview of the identified rearrangements and their comparison with FISH diagnostics is provided in Table 1. Across 20 control samples, FFPE-TLC detected no rearrangements, demonstrating the robust capability of PLIER in masking the intrinsic topological and methodological noise that inevitably is present in (FFPE) proximity-ligation datasets, while able to detect rearrangements involving *MYC, BCL2,* and *BCL6* across

the lymphoma samples. In total, PLIER identified 137 rearrangements involving *MYC, BCL2,* and *BCL6*: 56 *MYC* rearrangements (in 49 lymphoma samples), 39 *BCL2* rearrangements (in 34 samples), and 42 *BCL6* rearrangements (in 40 samples) (Fig. 2A).

**Distinguishing target fusions from unrelated chromosomal rearrangements**. To unambiguously assess whether PLIER-identified genomic regions were true rearrangements of the interrogated target genes, we closely inspected the distributions of their proximity-ligation products along with the linear sequences of each presumed partner, in so-called butterfly plots[26]. If engaged in a reciprocal translocation, each locus should reveal a "breakpoint" location separating its upstream sequences that preferentially form proximity-ligation products with one side of the partner locus, from its downstream sequences that preferentially contact and ligate the other part of the partner locus (Fig. 2B). Figure 2C shows three examples of reciprocal rearrangements uncovered by butterfly plots, involving *MYC, BCL2,* and *BCL6*, respectively. Rearrangements can also be non-reciprocal, such that only one part of a target locus fuses to a given partner. Fig. 2D shows butterfly plots of these more complex rearrangements of *MYC, BCL2,* and *BCL6*. Across all analyzed samples, *MYC* was found to be involved in 41 reciprocal translocations (26 with *IGH* and 15 with non-IG loci) and 15 more complex rearrangements (4 with *IGH*), *BCL2* in 34 reciprocal translocations (33 with *IGH* and 1 with *IGK*) and 5 more complex rearrangements, and *BCL6* in 37 reciprocal translocations (16 with *IGH*, 5 with *IGL* and 16 with non-IG loci) and 5 more complex rearrangements (Suppl. Figs. 3–5).

In addition to the 137 rearrangements with breakpoints in the *MYC, BCL2,* or *BCL6* locus, PLIER was expected to also detect two bystander categories of genomic rearrangements that also can yield significant enrichment in proximity-ligation products. The

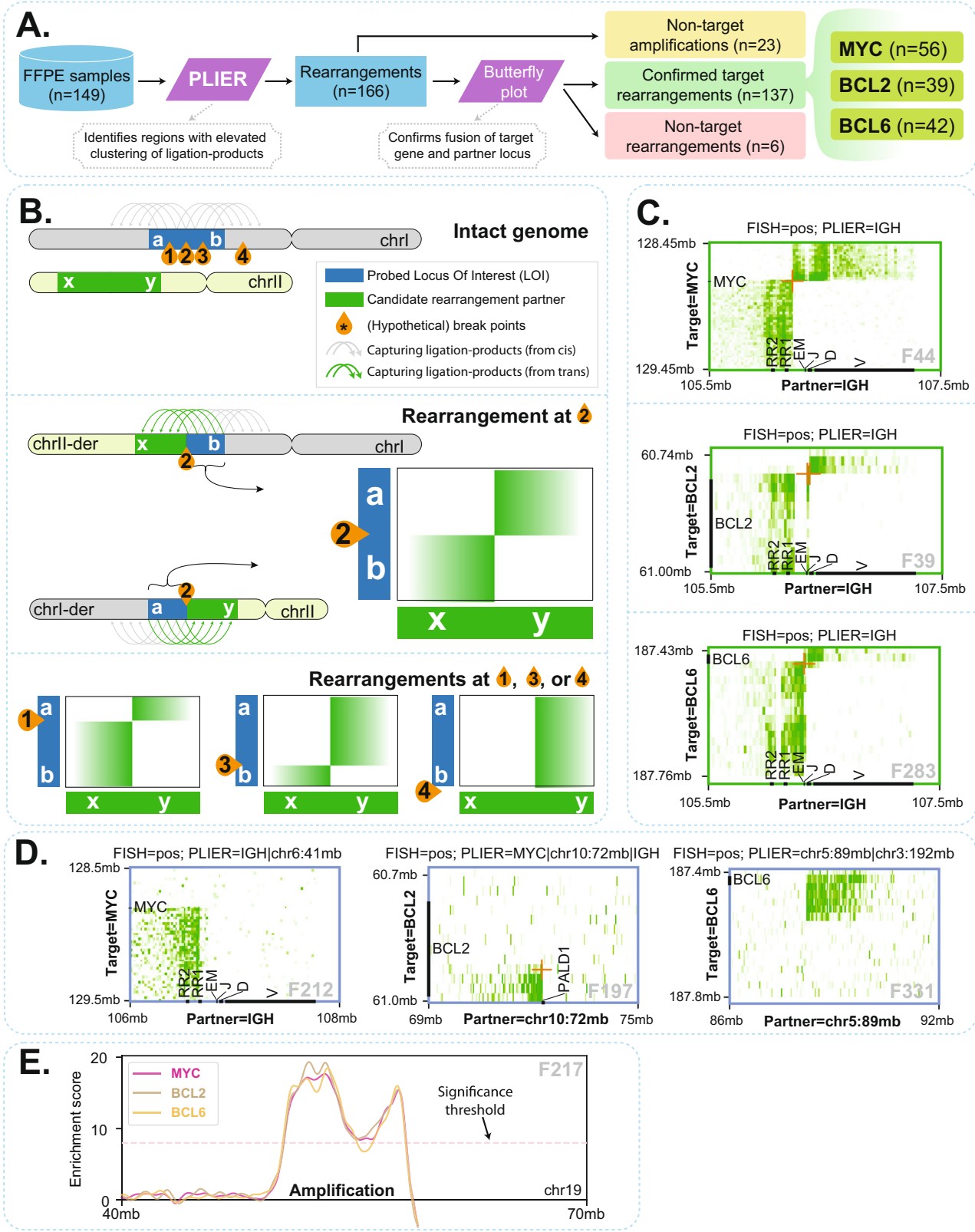

first was gained or amplified genomic regions (copy number variations); they could be distinguished from true positive rearrangements since PLIER scored them with all target genes (Fig. 2E). PLIER discovered 23 amplifications throughout the genome across all analyzed lymphoma samples. The second bystander category scored by PLIER were genomic rearrangements involving the chromosome that contained the target gene,

but with breakpoints outside the probe-targeted region. As a consequence, such rearrangement showed no linear transition in proximity-ligation signals between the identified rearrangement and the target locus in butterfly plots (see Fig. 2B). Six of these rearrangements were found and for two cases (F209 and F262) we confirmed a rearrangement involving chromosome 3 but with a breakpoint megabases away from the *BCL6* locus (Suppl. Fig. 6).

**Fig. 2 PLIER identified rearrangements. A** Overview of structural variant identification by PLIER. **B** Schematic explanation of how butterfly plots of proximity-ligation products (green arches on top of chromosomes) between the target gene and the PLIER-identified rearrangement partner can help distinguish true target rearrangements (breakpoints 1–3, inside the probe targeted region) from non-target rearrangements (breakpoint 4, outside the probe targeted region). In a reciprocal rearrangement inside the target locus, the locus should reveal a 5′ part (section a) that preferentially forms proximity-ligation products with one side of the partner locus and that separates from a 3′ part (section b) that preferentially contacts and ligates the other part of the partner locus. If a breakpoint is present in cis outside the probe-targeted region (breakpoint 4), a 5′ (a) and 3′ (b) part of the target gene cannot be distinguished. **C** Three examples of reciprocal rearrangements uncovered by butterfly plots, involving MYC, BCL2, and BCL6, respectively. **D** Rearrangements can be non-reciprocal, such that only one part of a target locus fuses to a partner, as exemplified using butterfly plots of MYC, BCL2, and BCL6. **E** An example of identified amplification events. Such events are apparent from the elevated number of ligation products that are captured by all target genes (shown for MYC, BCL2, and BCL6 genes).

Bystander rearrangements scored by PLIER were considered irrelevant for the gene of interest and were therefore classified as negative (Supplementary Data 2).

**FFPE-TLC uncovers known and previously uncharacterized complex rearrangements.** A graphical overview of the rearrangement partners identified in this study using Circos plots[27] is provided in Fig. 3A. In our collection of 149 samples, we found 3 samples positive for translocation in *MYC* and *BCL2* and *BCL6* (i.e., triple hit), 19 samples positive for translocation in both *MYC* and *BCL2* or *BCL6* (double hit), and 8 samples carrying a rearrangement in both *BCL2* and *BCL6* (see Supplementary Data 2). In 5 tumors, *MYC* was either directly fused to the *BCL6* (F72, F190, F194) locus, or involved in a complex 3-way fusion with *IGH* and *BCL2* (F197, F274). Apart from the immunoglobulin loci, we found several other recurrent rearrangement partners, including the *KYNU/TEX41* locus (F67, F188, with *BCL6* and F201 with *MYC*), *TBL1XR1* (F49, F273, F329, with *BCL6*), *IKZF1* (F210, F281, with *BCL6*) and the *TOX* locus (F74, F271, with *MYC*). Strikingly, *GRHPR* was found 5 times as a rearrangement partner of *BCL6* (F77, F199) and *MYC* (F202, F209, F269) (Fig. 3A). In cases such as F197 (*MYC*) and F331 (*BCL6*) we found strong indications for a non-reciprocal translocation event that fuses the different parts of the target locus to different genomic partners (Fig. 3B). In other instances, there was evidence for allelic three-way rearrangements, often involving the *IGH* locus, *MYC* (F50, F212, F274), *BCL2* (F193, F274, F282), or *BCL6* (F77) and a third partner (Fig. 3C, for examples). Further, in rare cases such as F67 (*BCL6*) (Fig. 3D), F202 (*MYC*), and F197 (*BCL2*) both alleles of the targeted locus independently appeared to be involved in rearrangements.

Using FFPE-TLC and PLIER, we were readily able to retrieve 90 breakpoint-spanning fusion-reads for the 137 identified SVs involving *BCL2*, *BCL6,* or *MYC* (Supplementary Data 3). Mapping the breakpoints to the target genes as well as to the *IGH* locus allowed inspection of recurrent breakpoint clusters in *MYC*, *BCL2, BCL6*, and *IGH*, as described previously[8,28] (Fig. 3E and Suppl. Fig. 7).

Even though probe design at IG loci was not optimal (as probes centered only on the enhancer regions), PLIER identified most (79 out of 91) rearrangements with *MYC*, *BCL2* and *BCL6* also reciprocally, when targeting the IG genes. Additionally, many rearrangements were found joining the IG loci with other genes, most of which have been described as rearrangement partners: IGH-PAX5/GRHPR (F21)[25,29] IGH-FOXP1 (F41)[30], IGH-PRDM6 (F43), IGH-CPT1A (F58)[31], IGL-BACH2 (F223)[32], and IGH-ACSF3 (F278)[33]. Such cases warrant further investigation, particularly since they were found in samples not carrying other known drivers of lymphoma (Supplementary Data 2).

**FFPE-TLC validation and sensitivity evaluation.** To further evaluate the robustness of our approach, we included a full

technical replicate (F49 and F68), twelve technical replicate samples for library preparation, capture, sequencing and PLIER and two technical replicate samples for capture, sequencing, and PLIER. In all instances, the exact same partners of *MYC, BCL2,* and *BCL6* were scored, even with remarkably similar z-scores (see Supplementary Data 2). Also, in samples F16 and F57 an apparently identical rearrangement was found. After inquiry, this appeared to be material taken in 2017 and 2018 from the same patient. For further validation and to explore alternative proximity-ligation methods, we processed six lymphoma samples by Hi-C. Despite much deeper sequencing (257M–540M Hi–C read pairs, compared to 17M–71M read pairs sequenced for FFPE-TLC), Hi–C failed to detect the known rearrangements, since the number of captured ligation-products at the rearrangement site was very limited (Suppl. Fig. 8). We then processed 47 FFPE samples with 4C-seq[34]. In 4C-seq, inverse PCR instead of hybridization capture is used to enrich proximity-ligation products that are formed with selected sites of interest[35]. For this study, a multiplex 4C PCR was used with 14 primer sets distributed over the *MYC*, *BCL2* and *BCL6* locus and 7 primer sets targeting the *IGH*, *IGL* and *IGK* loci (total 21 primer sets, see Suppl. Table 1). A modified version of PLIER was used to support the FFPE-4C type of data and score rearrangement partners (see Methods). Across all tested samples results were concordant between FFPE-TLC and FFPE-4C (Suppl. Table 2), with two exceptions (F54 and F67) where FFPE-4C failed to detect the rearrangement. Both were older samples, dating from 2007 and 2009, respectively, with severe DNA fragmentation. This suggested that FFPE-TLC is more tolerant to poor sample quality than FFPE-4C, which could be expected given that 4C additionally requires the circularization of (small) proximity-ligation products.

A major aim of our studies was to compare FFPE-TLC to FISH as a diagnostic method for rearrangement detection in FFPE specimens. Given background scoring results in negative control tissue, FISH is generally considered negative (i.e., no rearrangement is identified) in diagnostic practice if aberrant signals occur in less than 10–20% of cells (the exact cut-off can differ per gene and per diagnostic center). The sensitivity of FFPE-TLC relies on PLIER's ability to distinguish candidate rearrangement partners from the background noise. For all three target genes, we found somewhat higher enrichment scores for the immunoglobulin than the non-IG rearrangement partners (Suppl. Fig. 9 and Supplementary Data 2), presumably because our probe design also targeted (and enriched for) the IG loci. Further, *MYC* rearrangements less often received extreme (>60) enrichment scores, which is probably because we probed a much larger window around *MYC* (>1 Mb) than around *BCL2* and *BCL6* (260–330 Kb): with increased distance to the breakpoint the rearrangement signal is expected to diffuse. To more systematically investigate PLIER performance and sensitivity, we took six FFPE samples carrying FISH-validated rearrangements in *MYC* (2x), *BCL2* (2x), and *BCL6* (2x) with known percentages of FISH-positive cells, and

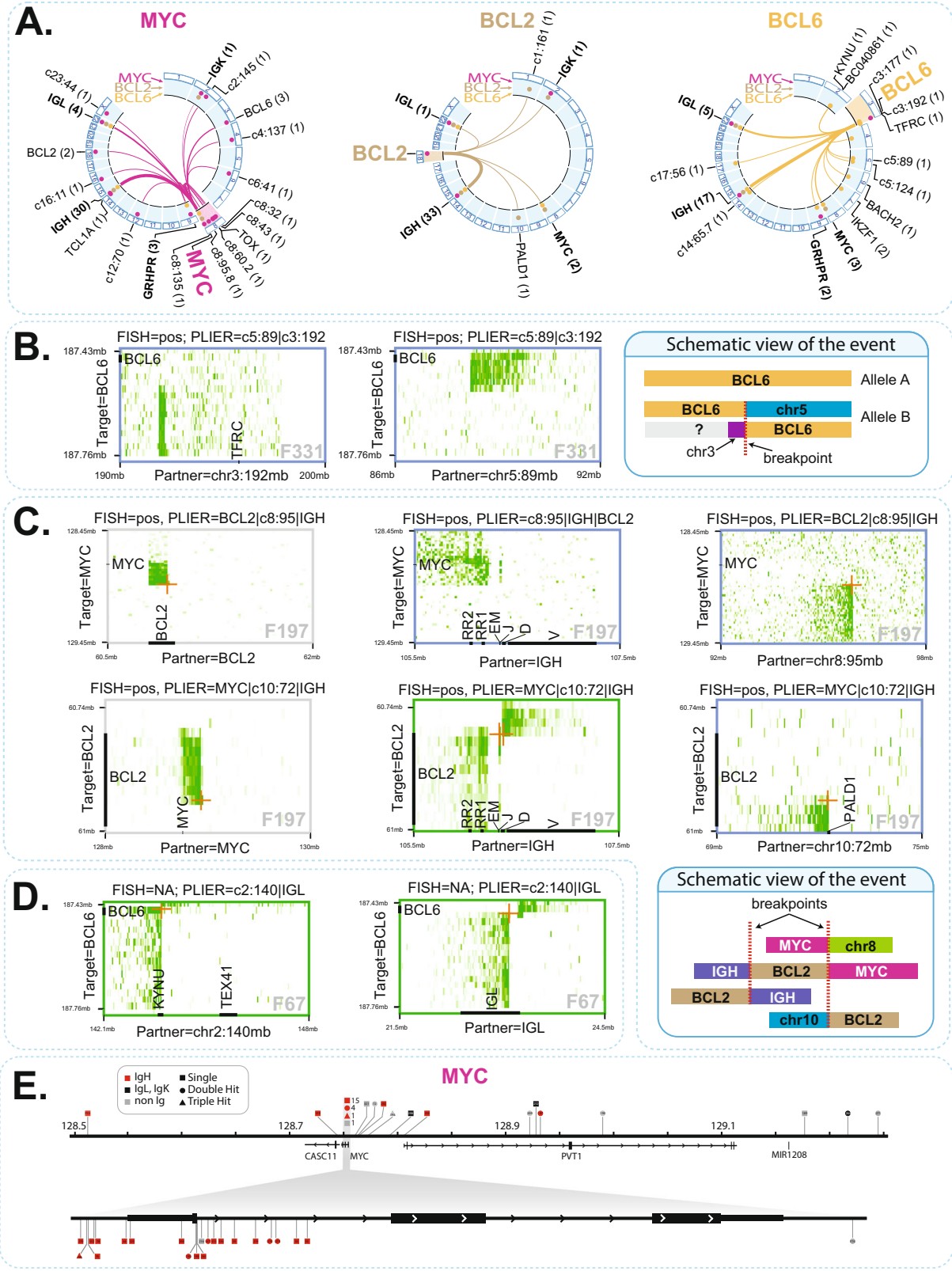

diluted each sample (prior to probe pulldown) with control material not carrying the rearrangement, to percentages of 5%, 1%, and 0.2%. As expected, we observed reduction of proximity-ligation products captured from the partner region (Fig. 4A). We found that PLIER identified the actual rearrangement partner in all samples having 5% or more rearranged cells (see Fig. 4B and Suppl. Table 3). Also, PLIER made no false-positive calls in any of

the diluted samples, which demonstrated the powerful statistical framework of PLIER in rejecting the intrinsic noise of FFPE-TLC datasets and only calling the true rearrangements. To estimate the minimum number of (on-target mapped) reads required to successfully identify the rearrangement partners, we in silico downsampled (by random draw) the datasets of the same six samples, before and after their dilution to 5% of rearranged cells.

**Fig. 3 Butterfly plots can identify varied types of rearrangements. A** Circos plots showing the rearrangement partners identified in this study, for translocations with MYC (pink), BCL2 (brown) and BCL6 (orange). Partners found by more than one target gene are indicated in bold. The frequency at which a given partner is found in our study is indicated in parentheses. Additionally, over the circumference of each Circos plot (highlighted in light blue), dots indicate the target genes (i.e., MYC with pink dots, BCL2 with brown dots, BCL6 with orange dots) that are found to be rearranged with each partner in our study. **B** Example of a non-reciprocal translocation event that fused the different parts of BCL6 to different genomic partners (chr3 and chr5). **C** Example of a complex, three-way rearrangement involving IGH, MYC, BCL2 as well as regions on chr8 and chr10, shown in butterfly plots as well as schematically. **D** An example in which both alleles of BCL6 are independently involved in rearrangements. **E** Overview of breakpoint positions identified in the MYC locus in our study. Such breakpoints are discerned in base pair resolution by mapping fusion-reads captured by FFPE-TLC.

We repeated this procedure 20 times, and each time we asked whether PLIER would call the known rearrangement. As shown in Fig. 4C, in the undiluted tumor samples not more than 75 K on-target reads were needed to robustly detect the *MYC*, *BCL2,* and *BCL6* rearrangements. When present in only 5% of the cells, one million on-target reads were sufficient for their detection. Collectively, our analyses showed that FFPE-TLC offers superior sensitivity when compared to FISH. However, the clinical implications of low rearrangement percentages caused by low tumor cell percentage or by tumor heterogeneity remain to be determined.

We compared the original FISH results to our FFPE-TLC results. Out of the 49 samples scored *MYC* positive by FFPE-TLC, 47 samples were also classified as such by FISH (Table 1), while two of these *MYC* rearrangements were missed by FISH. They were both rearrangements in *cis*, with partners on the same chromosome 8 (F16 and F221: here FISH detected multiple *MYC* signals (gain)) (Fig. 4D). For *BCL2*, 31 out of the 34 samples that we scored positive had also previously been reported by FISH: the three previously uncharacterized identified rearrangements, each carrying a *BCL2-IGH* translocation, had not been analyzed by FISH. For *BCL6*, 29 out of the 40 tumors with a *BCL6* rearrangement had also been scored as such by FISH. Three *BCL6* rearrangements (F38, F40, F49) were not detected by FISH (Fig. 4E), in two instances because of below threshold percentages of cells with a rearrangement (10% (F38) and 6% (F40)). In the third case (F49), FFPE-TLC detected a 1.35 Mb insertion of the TBL1XR1 locus into the BCL6 locus (Fig. 4F). With hindsight, some split of signals could be observed in the FISH image (Fig. 4G) that originally was considered irrelevant. Two FFPE-TLC identified *BCL6* rearrangements (one of which with *IGH*) were previously considered inconclusive by FISH because of single fluorescent signals (F25, F261). Six previously uncharacterized identified *BCL6* rearrangements (2x *IGH*, 2x *IGL*) had not been analyzed by FISH (Table 1). Vice versa, all rearrangements scored by FISH were confirmed by FFPE-TLC, except for two (F217 and F322, both described as having a complex karyotype). Whether FFPE-TLC or FISH was wrong here could not be determined, unfortunately. In summary, all 149 samples analyzed FFPE-TLC showed very high concordance with FISH. It missed two rearrangements scored by FISH but also identified and characterized two *MYC* rearrangements and five *BCL6* rearrangements that were not scored by FISH. Moreover, FFPE-TLC's capacity to analyze multiple genes in parallel for their involvement in rearrangements, enabled discovering 9 cases of *BCL2* and *BCL6* rearrangements in samples that had not been tested for these rearrangements by FISH. In four cases, this discovery changed the classification of the samples. Sample F16 could now be classified as "double-hit" (DH) for *MYC* and *BCL2* rearrangements, sample F67 as a *MYC* and *BCL6* DH tumor (with partners *IGH* and *IGL*), sample F194 as *MYC* and *BCL2* and *BCL6* triple hit (TH, although *MYC* and *BCL6* fused together) and sample F209 as TH.

We also wished to compare FFPE-TLC to the targeted DNA capture-based sequencing methods (Capture-NGS) for the

detection and analysis of structural variants in FFPE specimens[8–10]. For this, we compared Capture-NGS and FFPE-TLC performance on 19 FFPE samples that were part of a larger cohort of >200 FFPE samples previously analyzed by Capture-NGS. The selected 19 samples included a few samples in which the Capture-NGS results were discordant with the original FISH diagnoses. Fig. 5A shows the outcome of this comparison where 7 out of 7 translocations (from 6 lymphoma samples) in which Capture-NGS had failed to identify FISH-reported translocations were confirmed by FFPE-TLC (samples: F190 [*MYC* and *BCL6*], F197 [*MYC*] and F198 [*MYC*], F193 [*BCL2*], F188 [*BCL6*], F191 [*BCL6*], F192 [*BCL6*]). In four of these cases, the actual breakpoint was found outside the Capture-NGS probe targeted regions (F188, F197, F192, and F190 [*BCL6*]). Particularly in one case (F190), FFPE-TLC demonstrated that the *MYC* and *BCL6* rearrangements identified by FISH were actually a single *MYC-BCL6* translocation. Capture-NGS failed to find a breakpoint fusion-read and therefore missed this rearrangement because the *BCL6* breakpoint located outside the probe targeted region. Meanwhile no coverage was observed around the *MYC* breakpoint using Capture-NGS (Fig. 5B, left plot). Nonetheless, FFPE-TLC captured many ligation-products surrounding the breakpoint on both MYC and BCL6 sides (Fig. 5B, right plot). Thus, in cases where breakpoints occurred outside the probe-covered region, Capture-NGS failed to identify the rearrangement, whereas FFPE-TLC, as discussed, has no problem detecting such rearrangements. To illustrate this further, we reanalyzed datasets of six samples carrying a FISH-confirmed rearrangement with either *BCL2* (2x), *BCL6* (2x), or *MYC* (2x), but filtered the reads to exclusively consider ligation products that were captured made by a 50 kb interval of probes placed at increasing distance from the mapped breakpoint. Compellingly, in all instances, PLIER found the rearrangement with very high confidence (Fig. 5C). In three other cases (F191, F192, F198) Capture-NGS was not able to identify the rearrangement partner as the breakpoint has occurred at a non-unique sequence, whereas FFPE-TLC readily scored them (z-scores > 60). To further assess the difficulty that NGS strategies (which rely on breakpoint fusion-read mapping) have in identifying such rearrangements, we analyzed the mappability of all breakpoint-flanking sequences found in this study ($n = 347$), across different read lengths. Fig. 5D shows that around 5% of FFPE-TLC identified rearrangements would be missed (i.e., not be uniquely mappable) even when reading 60 nucleotides into the partner sequence. Finally, there was one case (F189) for which Capture-NGS identified fusion-reads suggesting a MYC translocation, which was unconfirmed by FISH as well as by MYC immunohistochemistry, and also FFPE-TLC did not identify the translocation. Detailed further analysis by PCR and sequencing revealed that this rearrangement was a small insertion placing 240 base pair of chromosome 8 into chromosome X, but not affecting the *MYC* locus (Fig. 5E).

In conclusion, FFPE-TLC offers clear conceptual advantages over regular capture-NGS methods for the detection of chromosomal rearrangements. Capture-NGS relies on breakpoint fusion-read identification for the detection of rearrangements,

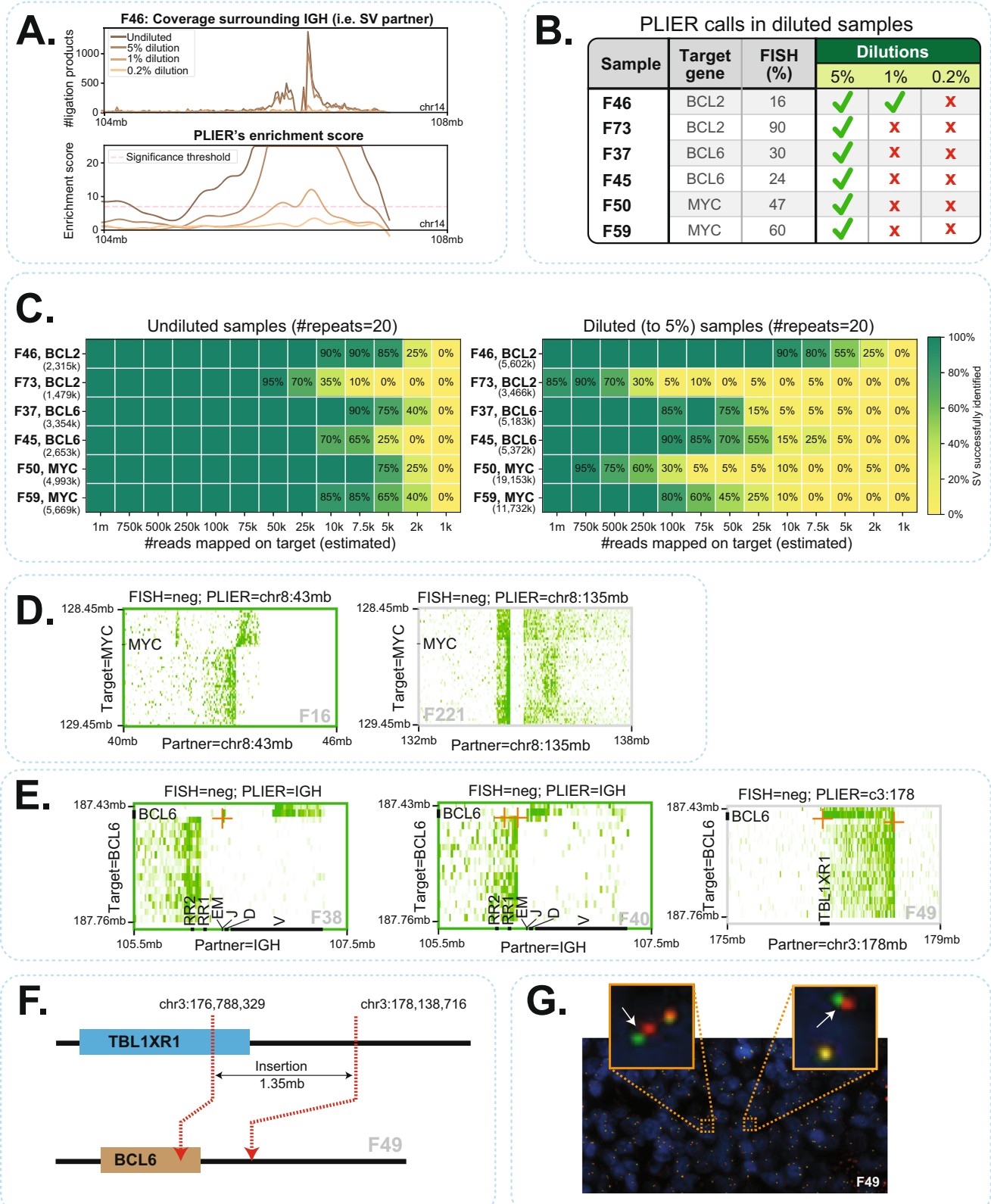

which is severely hampered when breaks occur outside the probe-covered region and/or in repetitive DNA. FFPE-TLC, as we show, accurately finds these rearrangements because it analyzes the proximity-ligation pairs between a target gene and its rearrangement partner.

## Discussion

We present here FFPE-TLC, a proximity-ligation-based method for targeted identification of chromosomal rearrangements in clinically relevant genes in FFPE tumor samples. As an assay to be applied in the diagnostic setting, FFPE-TLC offers important

**Fig. 4 Sensitivity and specificity of PLIER. A** Visualization of ligation products as well as PLIER-computed enrichment scores across dilutions for sample F46 that harbors a BCL2-IGH rearrangement. **B** Overview of PLIER identified rearrangements in diluted samples. Green checkmarks indicate successful identification of translocations by PLIER without any false-positive calls across the genome. Red crosses indicate failure of PLIER in detecting the rearrangement, either by missing the rearrangement or because of false-positive calls on other regions. **C** Downsampling analyses performed across diluted samples and their undiluted counterparts. The number of times PLIER successfully identified the rearrangement is reported as a percentage (out of 20 repeats). Any false-positive call by PLIER is considered as a failed identification of the rearrangement in that repeat. The total number of on-target reads mapped (i.e., without downsampling) is mentioned in parentheses under the sample identifiers. **D** Butterfly visualization of F16 and F221 that were negative for breaks in MYC by FISH. FFPE-TLC revealed that they in fact harbor a MYC rearrangement within the same chromosome. **E** Butterfly visualization of three BCL6 rearrangements (F38, F40, F49) that were missed by FISH. In two instances (F38, F40), FISH failed to identify the rearrangements as the percentages of cells with breaks were below threshold. **F** In F49, FFPE-TLC revealed that a 1.35 Mb section of the TBL1XR1 locus was inserted into the BCL6 locus. **G** BCL6 FISH image of F46 showing no breaks at initial inspection. With hindsight, the zoomed-in view (orange boxes) reveals some split signals (white arrows), but not above threshold (2 signal distances apart).

advantages over FISH, the current gold standard for targeted rearrangement detection in lymphoma FFPE samples. Firstly, unlike FFPE-TLC, FISH is highly dependent on good quality tissue and cell morphology, which may be negatively impacted by necrosis, apoptosis, and crush artifacts in resection specimens and by very limited material from core needle biopsy samples. We included core needle biopsy samples in this study, which showed that even very small samples yielded good quality FFPE-TLC results. No major differences in sensitivity and specificity were found between the FFPE samples provided by the five different clinical centers, showing that FFPE-TLC is resistant to the differences that may exist between their protocols for FFPE preparation and storage. Also, FFPE-TLC performed similarly on recent and older tissue blocks (Suppl. Fig. 10). Secondly, FISH results may give inconclusive results or lead to subjective interpretation in cases where aberrant numbers of FISH signals are seen per cell; FFPE-TLC offers the great benefit of objectively scoring rearrangements involving the selected target gene loci, based on a data analysis algorithm, PLIER. Thirdly, FFPE-TLC results provide much more detailed information on the rearrangement: not only does the method score whether or not the clinically relevant genes are intact or rearranged, as does FISH, it additionally identifies the rearrangement partner, the position of the breaks in relation to the genes involved, and, often, the fusion-read that describes the rearrangement at base-pair resolution. Collecting this detailed information in relation to disease progression and treatment response is anticipated to improve diagnosis, prognosis, and treatment of cancer patients. Translocation information at base-pair level also provides an individualized tumor marker to enable the design of tumor-specific personalized assays for minimal residual disease testing. Finally, FFPE-TLC is more sensitive: to avoid false positive calling, FISH assessment generally uses a 10–20% cut point of aberrant signals as set by a normal control reference and caused by "cutting off" signals from 10 to 20 μm diameter tumor cells in 3–5 μm sections. FFPE-TLC reliably detects rearrangements even if present in only 5% of the cells, which makes it also an interesting method to apply to fusion gene detection in solid tumors.

Whole genome sequencing (WGS) and regular NGS-capture methods are also used to identify SVs, find fusion partners and provide detailed information on the rearrangement breakpoint. WGS is however too expensive and computationally too demanding for a tool to diagnose rearrangements in selected target genes. Also, compared to these methods FFPE-TLC offers important advantages, particularly because it is not strictly reliant on (successful pulldown and) recognition of fusion reads. Rather, FFPE-TLC measures accumulated proximity-ligation events between chromosomal intervals flanking the breakpoint to identify a rearrangement. This, as we show, enables robust detection of rearrangements missed by regular NGS-capture methods, for example in cases when probes are not positioned

close enough to the breakpoint for pulling down the fusion read, or when non-unique sequences flanking the breakpoint compromise fusion-read recognition. In this study, we targeted genomic intervals of respectively 260 Kb, 330 Kb, and 1.05 Mb around the BCL2, BCL6, and MYC genes, i.e., regions that span previously identified rearrangement breakpoints in lymphoma[8,28]. A tiled probe design was used, but for selective pulldown of proximity ligated products probes may also be designed to only flank the (NlaIII) restriction enzyme recognition sites of interest[36]. In general, for FFPE-TLC, we recommend having probes at all restriction sites across the entire gene or locus of interest, plus at least 20Kb of its flanking sequences. As explained, by having sufficient proximity ligation information from flanking sequences, butterfly plots enable to unambiguously determine whether PLIER-identified chromosomal regions represent rearrangement partners fused directly to sequences inside the gene or locus of interest.

A critical aspect of our study was the development of PLIER, our computational/statistical pipeline to objectively interrogate a FFPE-TLC dataset for rearrangement partners. Currently utilized fusion-read finders that process data produced from targeted NGS approaches often require a certain level of manual data curation, precluding fully automated and parallel data processing. In FFPE-TLC, PLIER enables automated identification of chromosomal rearrangements, from processing of sequenced FFPE-TLC libraries to the delivery of simple tables that include identified rearrangements. PLIER searches within each test sample for chromosomal intervals with significantly enriched densities of independently ligated fragments, without the need for comparison to a reference (or control) dataset. It thereby accounts for differences in the intrinsic signal to noise levels across samples, which is essential given the relatively large range of DNA quality from FFPE samples from different tissues, different hospitals and different archival storage times and conditions. Initially trained on a curated dataset of 6 samples and then applied to the full dataset of all samples, PLIER demonstrates to be very robust against varying levels of noise, and at the same time sensitive in detecting rearrangements across all 149 samples in our study.

A large number of rearrangements in malignant lymphomas that were uncovered in this study warrant consideration in light of the World Health Organization (WHO) classification of lymphomas. Currently, aggressive B-cell lymphomas with a combined *MYC*- and *BCL2* and/or *BCL6* translocations (so-called double-hit or triple-hit, DH/TH lymphomas) are classified as a separate entity, irrespective of morphological features. The rationale for this is not only found in the aim for "biologically meaningful classification", but also in the characteristic poor clinical outcome that justifies a more intensified first-line treatment. More recently, in a very large series of such lymphomas, the Lunenburg Lymphoma Biomarker Consortium could show that this poor outcome is actually restricted to DH/TH

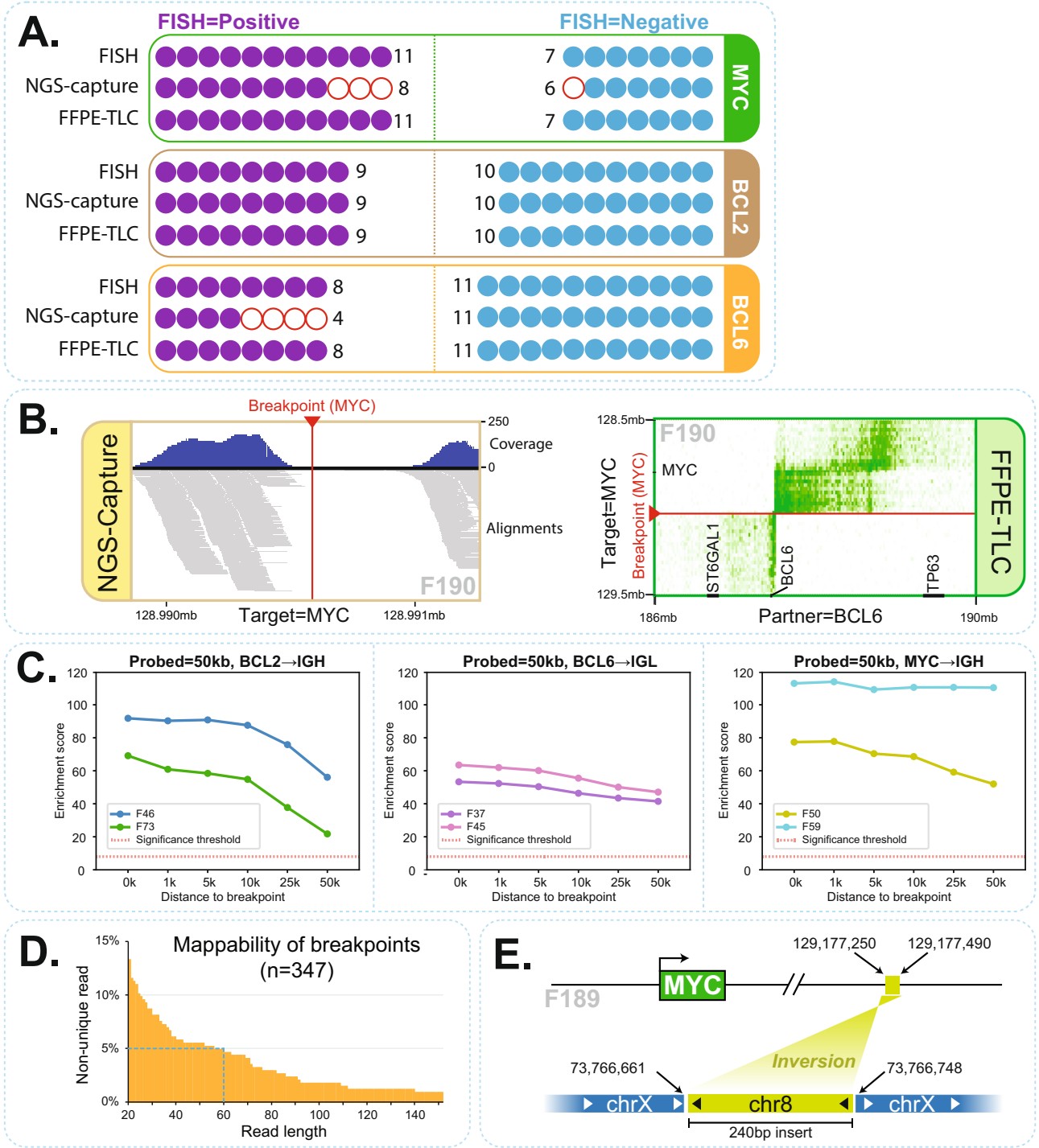

lymphomas with an IG-partner to the *MYC* rearrangement, while all other contexts (MYC-single hit, non-IG partners) have a similar outcome to DLBCL without a *MYC* rearrangement[37]. As a consequence, in the near future pathologists will be required to provide translocation status in aggressive B-cell lymphomas at this level of detail to support treatment decisions. Using FISH, 4 separate assays (*BCL2*,-BA (break-apart), *BCL6*-BA, *MYC*-BA, *MYC-IGH*-F(fusion)) are needed to diagnose DH/TH lymphomas, while still missing those cases that carry a *MYC-IGL* translocation since no commercial probes are available for *MYC-IGL* fusion FISH. Using FFPE-TLC, also this translocation context is diagnosed reliably in a single assay, which obviously improves time- and cost-effectiveness. We identified 4 cases with

*MYC-IGL* and one with *MYC-IGK*, of which one DH case (F264) in which clinical consequences would be immediate. We noted three cases of *MYC-BCL6* fusion (F072, F190, F194) and two cases fusing *MYC, BCL2,* and *IGH* (F197, F274) that by FISH would not be identified as such and interpreted as a DH context in four cases and TH context in one. It is unknown, however, if a single translocation event activates both translocation partner genes and results in a similar biological impact as two separate events. Similarly, both *MYC* and *BCL6* are frequently translocated to genes with a likely biological impact on malignant B-cell behavior (e.g., *TBL1XR1, CIITA, IKZF1, MEF2C, TCL1*). Nevertheless, until now the impact of such fusion partners could not be studied in clinical settings.

**Fig. 5 FFPE-TLC vs. other state of the art methods. A** Comparison of FISH, Capture-NGS and FFPE-TLC results showing rearrangements identified in MYC, BCL2, and BCL6 genes across 19 samples. Each circle is a sample that is analyzed for rearrangements in a particular gene. Filled-in circles indicate correspondence with FISH diagnosis and empty (red) circles indicate discordance with FISH diagnosis. **B** Example of false-negative call by Capture-NGS that was successfully identified by FFPE-TLC. It turned out that Capture-NGS had missed the rearrangement because the region around the breakpoint (red arrowhead) lacked coverage and therefore, the breakpoint could not be identified for sample F190. In contrast as shown in the butterfly plot, rearrangement identification by FFPE-TLC is fusion-read independent and therefore could correctly identify the rearrangement with high confidence (z-score = 82.4). **C** FFPE-TLC capabilities in detecting translocations even if breakpoints occur far away from the probed (targeted) regions. Each plot demonstrates this ability for a particular gene for two samples, from left to right: BCL2-IGH (shown for F46 and F73), BCL6-IGL (shown for F37 and F45), and MYC-IGH (shown for F50 and F59). The X-axis in each plot indicates the minimum distance between the last probe and the breakpoint position. The Y-axis shows enrichment scores that are computed by PLIER. In all tested cases, PLIER confidently identified the translocation even when the probes are located 50 kb away from the breakpoint. **D** Diagram showing the fraction of breakpoint sequences from this study that cannot be mapped uniquely on the reference sequence at varying read lengths. For example, even with 60 nucleotides, 5% of FFPE-TLC identified rearrangements would be missed by typical NGS capture methods due to unmappability of the captured sequence. **E** Schematic view of false-positive call by Capture-NGS in F189 sample. In this case, Capture-NGS identified reads that were spanning the breakpoint and linked the MYC locus to the X chromosome. In contrast, no rearrangement was identified by FFPE-TLC for sample F189. By performing PCR using primers on chromosome X and sequencing, we could successfully explain the event and confirm the insertion of a 240 bp fragment from chromosome 8 into chromosome X.

Since FFPE-TLC is based on regular capture protocols, we anticipate that FFPE-TLC analyses can also be designed to include the detection of clinically relevant SNVs and CNVs. This offers the possibility to develop methodology for the comprehensive diagnosis of all diagnostically relevant genetic variants.

In conclusion, FFPE-TLC combined with PLIER for objective rearrangement calling offers clear advantages over regular NGS-capture approaches and over FISH for the molecular diagnosis of lymphoma FFPE specimens. Future prospective studies should demonstrate how FFPE-TLC performs for other cancer types, like soft tissue sarcoma, prostate cancer and non-small cell lung carcinoma (NSCLC), which are also routinely screened in diagnostic pathology for the presence of clinically relevant chromosomal rearrangements in selected target genes. Following our design rules to have probes selectively positioned at all restriction enzyme recognition sites across a gene plus 20 kb of both of its flanking sequences, it should be feasible to include over 40 genes in a single probe panel, enabling simultaneous detection of their involvement in a chromosomal rearrangement. For additional detection of clinically relevant SNVs and mutations, the recommendation would be to include tiling probes across the exons of relevant target genes.

## Methods

**Patient samples.** This retrospective study used a set of 129 archival B-cell Non-Hodgkin lymphoma tissue samples, which were selected by the respective sites, and may therefore not represent an entirely random selection of samples in the respective sites. The corresponding lymphoma patients had been diagnosed between 2007 and 2019 at the University Medical Centre Utrecht, Amsterdam University Medical Centre—location VUMC, Laboratorium Pathologie Oost-Nederland, Leiden University Medical Centre and University Medical Centre Groningen and their affiliated hospitals. They had been mostly diagnosed as DLBCL, but also Burkitt, follicular and marginal zone lymphomas and some other diagnoses were included. 20 Non-lymphoma control samples were also analyzed, mostly reactive lymph node samples and tonsillectomy specimens. Formalin-fixed and paraffin-embedded (FFPE) tissue samples were obtained using standard diagnostic procedures. Per patient, 1 or more 10 μm scrolls or 4 μm unstained sections of the FFPE tissue blocks were provided for FFPE-TLC analysis in tubes or on slides.

The study was performed in accordance with the local institutional board requirements and all relevant ethical and privacy regulations were followed during this study. Informed consent was provided by the patients for the use of their tissue samples in this work. The use of tissue specimens and associated data in this study was approved by the Medical Ethical Committee of the University Medical Center Groningen (RR 201800551) for explorative research, Medical Ethical Committee of LabPON under "nader gebruik geen bezwaar", the TcBio of UMCU as "gebruik van restmateriaal", TcBio of VUMC/AUMC under "nader gebruik geen bezwaar" and the Medical Ethical Committee of LUMC under code of conduct of secondary use of tissues.

**Molecular analysis.** All patient samples had been analyzed with routine FISH with break-apart probes and fusion-probes in selected cases, in the majority of cases for

all 3 genes BCL2 (Cytocell LPS028; Vysis Abbott 05N51–020; IGH/BCL2 Dual Fusion Vysis Abbott 05J71–001), BCL6 (Cytocell LPH 035; Vysis Abbott 01N23-020) and MYC (Cytocell LPS 027; Vysis Abbott 05J91-001; IGH/MYC/CEP 8 Dual Fusion Vysis Abbott 04N10-020). A subset of 19 samples had also been analyzed with a Capture-NGS method as developed by the Amsterdam University Medical Centre – location VUMC team. A detailed description of this approach is provided in the Supplementary Materials & Methods.

**FFPE-TLC library preparation.** A step-by-step protocol to prepare FFPE-TLC libraries is provided in the Supplementary Materials & Methods. In brief, single FFPE sections were supplied by the medical centers in this study as scrolls in 1.5 ml vials or on slides. If a slide was provided, the contained material in the slide was scraped and transferred to a 1.5 ml vial. Excessive paraffin was removed by a 3-minute 80 °C heat treatment, followed by a centrifugation step after which the tissue was disrupted and homogenized by sonication using a M220 Focused-ultrasonicator (Covaris). Samples were primed for enzymatic digestion through incubation with 0.3% SDS for 2 h at 80 °C, then digested with NlaIII (a 4 base pair cutter restriction enzyme; NEB) at 37 °C for 1 h, and finally ligated at room temperature for 2 h with T4 DNA ligase (Roche). Next, a complete reverse crosslinking was done by overnight incubation at 80 °C and the DNA was purified using isopropanol precipitation and magnetic bead separation. Following elution, 100 ng of the prepared material was fragmented to 200–300 bp (M220 Focused-ultrasonicator, Covaris) and subjected to NGS library prep (Roche Kapa Hyper-prep, Kapa Unique Dual indexed adapter kit). A total of 16–20 independently prepared libraries were equimolar pooled with a total mass of 2 μg and subjected to hybridization with the capture probe pool, wash steps, and PCR amplification using the Roche Hypercap reagents and workflow according to the manufacturer's instructions. Paired-end sequencing was done on an Illumina Novaseq 6000 sequencing machine. All proximity-ligation libraries were sequenced deeper than deemed necessary (see Supplementary Data 2). The samples with lowest coverage were sequenced to a read depth of around 20 M, which invariably was sufficient for rearrangement detection.

**FFPE-TLC data processing (estimated duration: 12 h).** Sequenced reads from individual samples (i.e., patients) were mapped to the human genome (hg19) using BWA-MEM (version: 0.7.17-r1188; settings: -SP -k12 -A2 -B3) in paired-end mode[38]. BWA-MEM aligner allowed "split-mapping" in which a single read can be mapped into multiple fragments (i.e., separate regions) in the genome. This was essential to map FFPE-TLC data, as each sequenced read in FFPE-TLC may contain multiple fragments mapping to varied locations in the genome (see Suppl. Fig 1). Any fragments with mapping quality (MQ) above zero were considered as mapped, as is commonly done for proximity-ligation data processing[35,39]. Reads were assigned to their related target gene or "viewpoint" (i.e., a probe set such as MYC, BCL2, etc.) based on their fragment's overlap with the viewpoint's coordinates (see Supplementary Data 1 for probe set discordates). A read was discarded if it did not overlap with any viewpoint. In cases with fragments within a read that had overlap with multiple viewpoints, the read was assigned to the viewpoint with the largest overlap. As a result of this procedure, for each combination of sample and viewpoint, an independent FFPE-TLC alignment file (BAM) was produced.

The reference genome was split in silico into "segments" based on the recognition sequence of NlaIII restriction enzyme (CATG) where each segment starts and ends with an NlaIII recognition site. Mapped fragments were then overlaid on the segments. Due to rare alignment errors, more than one fragment within a read can overlap a segment. In such a case, only one fragment was counted for that particular segment and extra overlapping fragments on that read were ignored. We used HDF5 format[40] to store FFPE-TLC datasets which is a cross-

platform and cross-language file storage standard and therefore delivers convenience to future users of FFPE-TLC.

**Rearrangement identification by PLIER (estimated duration: 6 h).** In a given FFPE-TLC dataset, PLIER initially splits the reference genome into equally spaced genomic intervals (e.g., 5 kb or 75 kb bins) and then calculates for every interval a "proximity frequency" that is defined by the number of segments within that genomic interval that are covered by at least one fragment (i.e., a proximity-ligation product), see Suppl. Fig. 2 for a schematic overview of the entire procedure. "Proximity scores" are then calculated by Gaussian smoothing of proximity frequencies across each chromosome to remove very local and abrupt increase (or decrease) in proximity frequencies that are most likely spurious. Next, an expected (or average) proximity score and a corresponding standard deviation are estimated for genomic intervals with similar properties (e.g., genomic intervals present on trans chromosomes) by in silico shuffling of observed proximity frequencies across the genome followed by a Gaussian smoothing across each chromosome. Finally, a z-score is calculated for every genomic interval using its observed proximity score and the related expected and standard deviation of proximity scores. Finally, by combining z-scores calculated from multiple scales (i.e., interval widths such as 5 kb and 75 kb), a scale-invariant *enrichment score* is calculated (see Enrichment score estimation and Parameter optimization for PLIER sections for details). This scale-invariant enrichment score is used to recognize genomic intervals with elevated clustering of observed ligation products.

For genomic intervals present on cis chromosomes, we first corrected the known elevated proximity frequencies of genomic intervals adjacent to the targeted loci. To this end, for a given FFPE-TLC dataset we initially excluded the probed area as well as the surrounding ±250 kb area. Then, we performed a Gaussian smoothing (σ = 0.75, span = 31 intervals) on proximity frequencies on both sides of the probed area until the chromosome ends. Next, inspired by peakC[39], we performed an Isotonic-regression on the smoothed proximity frequencies. For each cis-interval we considered the difference between its smoothed proximity frequency and the corresponding Isotonic-regression prediction value as its proximity score. This procedure ensures that the known elevation of proximity scores in genomic intervals adjacent to the targeted (or probed) loci is accounted for. Finally, enrichment scores for cis intervals were calculated following a shuffling procedure similar to trans intervals (described above). We discarded cis-rearrangements identified in the ± 3 mb region around the viewpoint (i.e., closer than 3 mb to the viewpoint measured across the linear chromosome) to make sure the true 3D interactions between the viewpoint and its vicinity is not considered as rearrangement.

It is worth noting that the above statistical approach works well when a FFPE-TLC dataset is not sparse and is at least minimally populated with independent ligation products (i.e., coverage on diverse genomic segments in the genome). However, a sparse FFPE-TLC can arise from a library prepared with poor sample (tissue) quality, DNA extraction, low digestion or ligation efficiency, or other difficulties in library preparation. In such cases, only a minimal number of genomic intervals in the genome will have a proximity score above zero. As a result, the utilized permutation strategy (i.e., random shuffling of intervals) will underestimate the truly expected proximity score and therefore many intervals with proximity score above zero will be falsely considered as enriched. To remedy this issue, we considered a complementary permutation approach in which we only swapped the genomic intervals with proximity frequency above zero (instead of random shuffling of all intervals) and then calculated the corresponding z-scores by comparing the observed and expected proximity scores that are calculated using the swapping permutation strategy. For each genomic interval, we took the minimum z-score between the shuffling and swapping permutations as the final z-score for that particular genomic interval. This addition limited the number of false-positive calls even in a sparse FFPE-TLC dataset and makes PLIER suitable for FFPE-4C experiments as well. In all permutations, we repeated the shuffling or swapping 1000 times to estimate the corresponding expected and standard deviation of proximity scores.

It is important to note that in this approach, we do not correct for known biases such as GC content, mappability, segment, or restriction site density (i.e., number of restriction sites per interval) or a number of other known factors that could influence captured proximity frequencies. Owing to PLIERs flexibility, these parameters can be considered in the background estimation by only swapping (or shuffling) intervals that have similar chromatin compartment, GC content, restriction site density, etc. Nonetheless, our preliminary analyses did not show a considerable improvement when these parameters were corrected for in the background estimation and therefore, we opted for simplicity of the model which in turn reduces the computational demand of PLIER. This decision was especially important because we aimed to produce a light-weight pipeline that is suitable to be implemented in a clinical setting with minimal computational requirements.

**Enrichment score estimation (estimated duration: 2 h).** For a given sample (e.g., a patient) and viewpoint (e.g., BCL2) and genomic interval width (e.g., 5 kb), we initially selected genomic intervals that showed z-score above 5.0 and merged the neighbor selected intervals if they were closer than 1 mb. We took the 90-percentile z-score values of the merged intervals as their integrated z-score. To estimate the "scale-invariant" enrichment score from multiple interval widths (e.g., 5 kb and 75

kb), we grouped merged intervals that were closer than 10 mb and took the z-score value of the intervals with the largest scale (75 kb in this case) as the final enrichment score. Each collection of merged intervals across scales is referred to as a "call" in this study.

**Parameter optimization for PLIER (i.e., training phase).** To identify PLIER's optimal parameters, we used a collection of six FFPE-TLC samples, three lymphoma ("positive") and three control ("negative") samples. Specifically, three lymphoma samples (i.e., F73, F37, and F50) were included which, based on FISH (the gold standard), were expected to have a single rearrangement in BCL2, BCL6, or MYC, respectively while lacking rearrangement in the other two genes. The other three "negative" datasets (i.e., F29, F30, and F33) controlled datasets for which no rearrangements were expected in any of the three genes, see Supplementary Data 2 for details. We limited the optimization to BCL2, BCL6, and MYC genes as we only had clinical/diagnosis FISH data for these genes. We also included dilution (i.e., 5%, 1% and 0.2%) experiments of the three lymphoma samples (i.e., F73, F37 and F50) in the optimization procedure. Taken together, we had 12 positive cases (the 3 original patients, plus 3 additional dilution samples for each patient) for which PLIER should identify a rearrangement (i.e., "true positives" set) and 33 negative cases (3 control samples each with three genes, plus the two non-rearranged genes in 12 lymphoma samples) for which PLIER should not identify any rearrangement across the genome (i.e., "true negative" set). See Supplementary Data 2 for details on the included samples in the training phase. Apart from the correctly identified rearrangements, any extra rearrangement found in the positive cases across the genome were also considered as "false-positive" rearrangements. As a performance measure, we used the area under precision recall (AUC-PR) instead of Area Under the Curve as we potentially had more negative cases than positive cases (i.e., unbalanced class frequencies).

For an effective performance of PLIER's statistical framework, several parameters need to be optimally defined. We performed a massive parameter sweep using high performance computing (HPC) of University Medical Center Utrecht to identify the optimal parameters for PLIER. These parameters include: Gaussian smoothing degree (σ = 0.1, 0.25, 0.5, 0.75, 1.0, 1.5, 2.0, 2.5, 3.0, 3.5, 4.0), number of genomic intervals that the Gaussian kernel spans (#step = 11, 21, 31, 41, 51, 61) and genomic interval width (width = 5 kb, 10 kb, 25 kb, 50 kb, 62 kb, 75 kb, 100 kb). For interval widths, we also tested if combining multiple interval widths (i.e., scale-invariant enrichment scores) would perform better. Additionally, to identify how the z-scores of merged intervals (i.e., the intervals within 1 mb neighborhood of each other) should be integrated, we considered experimenting with maximum, 90th percentile and median operators. The measure of performance was chosen to be the area under precision-recall curve.

After the parameter sweep, we identified the followings as optimal parameters of PLIER: Gaussian smoothing σ = 0.75, Gaussian kernel span #step = 31, interval widths = 5 kb + 75 kb combined (but both z-scores should be above 5.0) and 90th percentile of z-scores of neighbor intervals (<1 mb) being merged as their final z-score. Finally, a significance threshold needed to be estimated to consider a call to be significantly enriched. By setting the maximum False Discovery Rate (FDR) as 1%, we reached significance of 8.0 as the optimal significance threshold for enrichment scores of trans-intervals (see Suppl. Fig. 11). Due to computational constraints and limited availability of diagnostic data, we only optimized PLIER parameters for trans-intervals of BCL2, BCL6, and MYC. We then used these parameters (without further optimization) for trans-intervals of other genes in the study (i.e., IGH, IGL, and IGK). For cis-intervals of all genes in our study, we again used the aforementioned parameters, with the exception of the significance threshold. For these calls, we took a conservative approach of much higher significance threshold (i.e., >16.0). Each output call from PLIER consisted of two genomic coordinates that indicate the boundary in which the scale-invariant enrichment score was above the significance threshold.

**Amplification detection (estimated duration: 1 h).** Although FFPE-TLC is not designed to identify amplifications, repeated rearrangements identified by PLIER from different probe sets but in the same sample and region can be indications of amplification events in that region. To leverage this prospect, we focused on the three primary genes in our study (i.e., MYC, BCL2, and BCL6) for which relatively large areas were probed (see Supplementary Data 1 for details). For each sample, we asked if a particular rearrangement (i.e., in the same region) is reported from more than one gene. An example of such amplification identified by PLIER is depicted in Fig. 2E. A complete list of the identified amplifications is provided in Supplementary Data 2. Of note, lymphoma samples could potentially harbor double hit rearrangements (e.g., BCL2 and MYC) specifically to the IGH area. To avoid calling such a rearrangement as amplification events, we excluded calls to the IGH area from amplification detection analysis.

**Blacklisted areas.** We noted that our IGL and IGK probe sets tend to repeatedly identify specific regions in the genome. We observed such calls even in our control samples for which no rearrangements were expected to be present. Specifically, our IGL probe set frequently identified chr9:131.5–132.5 mb and our IGK probe set frequently identified chr22:22–24 mb region of the human(hg19) genome. It is worth noting that the chr22:22–24 mb area harbors the IGL gene and therefore

such calls could potentially be interesting to investigate further. However, we noted that the corresponding *IGL* viewpoints did not identify *IGK* reciprocally. Consequently, we considered the elevation of enrichment scores to be due to a high sequence similarity between *IGL* and *IGK* that is likely to cause misalignments during the mapping procedure. Taken together, we considered both areas as off-target bindings of *IGK* and *IGL* probes, respectively, and ignored any rearrangements identified by these two probe sets in these areas.

**Fusion-read identification**. To identify fusion-reads in a given FFPE-TLC dataset (e.g., *MYC*), we collected split-alignments (i.e., individual read sequences that mapped to multiple areas in the genome). Then, the split-alignments that referred to enzymatic digestion in FFPE-TLC were filtered out by discarding the split-alignments that fused at a restriction enzyme recognition site in the genome (±1 base pair). The split-alignments that occurred at the rearranged coordinates (identified by PLIER) were manually checked in IGV to confirm the existence of read-fusions.

**Fusion-read mappability**. The identified breakpoint coordinates from the fusion reads were used in the mappability analysis to extract the corresponding sequences from the reference genome. In total 347 sequences of 151 bp (equal to the sequencing read length) upstream and downstream of the breakpoints were extracted from the reference genome. These 347 sequences were aligned using BLASTn (version: 2.8.1; settings: -perc_identity 80 -dust no -evalue 0.1) at different sequence lengths from 20 to 151, using a step size of 1 bp. The blast results were parsed to count the sequences with exact hits at each length; if exactly one hit, the sequence is considered unique, if multiple hits the sequence is considered non-unique. The fraction of non-unique sequences were plotted in a bar graph.

**Confirmation of the 240 bp chr8 insertion into chrX in sample F189**. A $2 \times 20$ cycles nested PCR was performed on control DNA and DNA isolated from sample F189 (Nebnext Q5 mix, NEB) using two primers for the initial PCR flanking the insertion on chrX (Fwd: ATTTTGATCGGCTTAGACCA, Rev: GGTTGATCAAAGCCAGTC) and 2 primers for the nested PCR (Fwd: GTCCAGCTTTGTCCTGTATT, Rev: GTCATGGCTGGTCAAGATAG). PCR products were separated on agarose gel, showing the expected sized product with insertion had been formed only for sample F189. For further confirmation the primary PCR products were amplified in the same nested PCR but now including Illumina sequencing adapters and an index sequence (Fwd: GTGACTGGAGTTCAGACGTGTGCTCTTCCGATCTGTCCAGCTTTGTCCTGTATT, Rev: ACACTCTTTCCCTACACGACGCTCTTCCGATCTGTCATGGCTGGTCAAGATAG) and subjected to sequencing (Illumina MiniSeq).

**FFPE-4C library preparation**. We processed 47 FFPE samples with 4C-seq[34]. In 4C-seq, inverse PCR instead of hybridization capture is used to enrich proximity-ligation products that are formed with selected sites of interest[35]. For this study, a multiplex 4 C PCR was used with 14 primer sets distributed over the *MYC, BCL2,* and *BCL6* locus and 7 primer sets targeting the *IGH, IGL* and *IGK* loci (total 21 primer sets, see Suppl. Table 1).

**HiC library preparation and data processing**. For Hi-C library preparation, FFPE samples were processed exactly as described for the FFPE-TLC targeted libraries (see FFPE-TLC library preparation), except that the probe hybridization and pulldown steps were omitted. The prepared HiC libraries (F50dil5, F59dil5, F209, F197, F199, F67) were sequenced in an Illumina NovaSeq 6000 machine in 2x150bp paired-end mode. The corresponding FASTQ files were processed following the 4DN HiC processing pipeline recommendations[41]. The resulting pairix files were accessed by Pairix[42] (v0.3.7) to produce the butterfly plots (shown in Suppl. Fig. 8) by visualizing the captured interactions between two regions in the genome (target vs. rearranged partner) in a heatmap similar to the standard HiC matrices. The bin width for each butterfly plot is chosen as 20 kb (or 50 kb if the plot was too sparse).

**Downsampling analyses**. We performed downsampling analysis by randomly drawing reads mapping to our target of interest (i.e., MYC, BCL2, or BCL6) that underwent a rearrangement. We did that across diluted samples and their undiluted counterparts and each downsampling experiment was repeated 20 times. The result of this analysis is shown in Fig. 4C. The number of times PLIER could successfully identify the rearrangement is reported as a percentage and shown over the heatmap (e.g., 90% refers to 18 successful identification of the rearrangement out of 20 repeated experiment). The percentage number is not shown if all repeats of the same experiment successfully identified the rearrangement (i.e., 100% success rate). Any false-positive call by PLIER is considered as a failed identification of the rearrangement in that repeat.

**Reporting summary**. Further information on research design is available in the Nature Research Reporting Summary linked to this article.

## Data availability
All sequencing data used in this study were mapped to the reference genome (hg19) and are available through the European Genome-phenome Archive (EGA study ID: EGAS00001004760). Of note, to protect patients' privacy, this submission is fully anonymized and is protected by the UMC Data Access Committee. Formal approval is needed to download the data.

## Code availability
PLIER's code[43] used in this manuscript is available at GitHub: https://github.com/deLaatLab/PLIER. This repository includes, in a step-by-step manner, how PLIER can be used to process FFPE-TLC and FFPE-4C sequenced samples. This repository also includes a simple "test" example which demonstrates a demo functionality of the entire procedure using PLIER. Additionally, our codebase is also available in Zenodo which contains the version of PLIER used in this study. Please refer to the following link to access this repository: https://zenodo.org/badge/latestdoi/300543907.

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

## Acknowledgements

We thank Cheryl Dambrot for the helpful comments on the manuscript. The work conducted by the Hubrecht Institute was supported by KWF/Alpe project 11632/2018-1; TKI project LSHM15017 (SuperGeneSeq); Oncode Institute. The work conducted by Cergentis has received funding from the European Union's Horizon 2020 SME instruments program SEQURE under grant agreement No. 806446.

## Author contributions

A.A. designed PLIER, data interpretation, wrote manuscript, prepared figures; M.P.: 4C-FFPE experiments, data interpretation, helped writing manuscript, prepared figures; J.S. optimized and executed FFPE-TLC experiments, data interpretation, helped writing manuscript, prepared figures; G.T.L.-d.V. selected samples, data interpretation, performed NGS-capture experiments, helped writing manuscript, helped prepare figures; M.Y. optimized FFPE-TLC experiments; R.L. selected samples, helped with data interpretation; R.W.J.M. selected samples, helped with data interpretation; R.vdG. selected samples, helped with data interpretation; J.V. selected samples, helped with data interpretation; A.C. selected samples, helped with data interpretation; T.vW. selected samples, helped with data interpretation; A.D. selected samples, helped with data interpretation; L.C.v.K. selected samples, helped with data interpretation; N.J.H. selected samples; P.S. performed NGS-capture wet-lab experiments, helped writing manuscript; M.Sh. helped with 4C-FFPE experiments; A.S.J.M. helped with rearrangement detection data analysis; P.J.P.d.V.: 4 C experiments to optimize rearrangement detection; M.J.A.M.V.: 4 C experiments to optimize rearrangement detection; P.H.L.K. helped with rearrangement detection data analysis; A.R. optimized FFPE-TLC experiments; K.H. designed FFPE-TLC data analysis tools, data interpretation, prepared figures; M.S. designed FFPE-TLC data analysis tools, data interpretation; M.v.M. helped conceive the study; B.Y. helped writing manuscript, helped with data interpretation, conceived and supervised capture-NGS study; D.d.J. helped writing manuscript, helped with data interpretation, conceived and supervised capture-NGS study; H.F.: helped writing manuscript, helped with data interpretation, conceived and supervised the study; E.S. conceived and supervised study, data interpretation, helped writing manuscript, prepared figures; W.dL. conceived and supervised study, data interpretation, wrote manuscript. M.P., J.S., and G.T.L.-d.V. contributed equally in this work.

## Competing interests

The authors declare the following competing interests: W.d.L., E.S., M.v.M. are founders and shareholder of Cergentis. J.S., M.Y., K.H., M.v.M., H.F., E.S. are employees of Cergentis. The remaining authors declare no competing interests. A list of patents and patent applications related to this work can be found in Suppl. Table 4.
