## [Peer Review File · Nature Communications]

Reviewers' Comments:

Reviewer #1:

Remarks to the Author:

This manuscript describes the feasibility of an enhanced FFPE-Targeted Locus Capture method for the detection of structural variants and in particular gene fusions in hematological disorders. Although a version of this method had already been published, this assay presents a number of important novelties, both in the experimental design (where a crucial gene enrichment step is added using capture probes) but also describes a new analysis pipeline "PLIER" that allows the detection and visualization of those events. Generally, for accurate gene fusion detection, routine capture-based methods require a comprehensive probe design that fully captures a genomic sequence where a fusion breakpoint is predicted to occur. This includes very long non-coding regions, hampering design important regions involving BCL2/6 and MYC. Therefore, detection methods such as FFPE-TLC/PLIER represent an important diagnostic tool in the clinical setting. The manuscript is well written and very detailed.

I have the following comments/suggestions to the authors:

- 1) It would be informative to include the duration of the PLIER analysis pipeline.
- 2) Line 191: "in our collection sample": please clarify whether you're referring to the 149-sample cohort.
- 3) Line 236: did you mean "false negative"?
- 4) Lines 242-243: "Out of the 49 samples scored MYC positive by FFPE-TLC, 47 samples were also classified as such by FISH". This statement is discordant with Table 1. According to the table, FFPE-TLC detects 47 MYC rearrangements and 2 are negative whereas 49 samples are positive by FISH.
- 5) Were the 2 MYC negative samples by FFPE-TLC tested by a different method (e.g. 4C-seq) to confirm that those weren't false negatives? No additional experiments are needed but simply include the data if available.
- 6) Based on the analysis performed in this paper, I don't believe that a general statement indicating that FFPE-TLC is superior our outperforms regular NGS capture-based sequencing is very accurate. The NGS assay that was used here wouldn't capture the fusion breakpoints due to a design limitation. Therefore, those fusions are expected to be missed. There are capture-based assays that include BCL2/6, MYC fusion design (e.g. FoundationOne Heme). If such an assay (or other without a known design limitation) fails to detect a gene fusion event that's picked up by FFPE-TLC, then you can say that your assay outperforms that specific NGS capture assay. I recommend softening the language around that in the introduction/conclusion.
- 7) In the discussion, please mention Whole Genome Sequencing as a very accurate method for the detection of all SVs but discuss the limitations (cost, computational requirements, etc).
- 8) There are few typos across the document (e.g. Figure 2B "HypotHetical")

Reviewer #2:

Remarks to the Author:

In the study "Robust detection of translocations in lymphoma FFPE samples using Targeted Locus Capture-based sequencing", Allahyar et. al., develop and validate FFPE-TLC - a proximity-ligation based method for targeted identification of chromosomal rearrangements in FFPE-fixed lymphoma samples. This method cleverly exploits the proximal fixing of DNA that occurs during FFPE fixation in sample preparation to identify chromosomal translocations that are a cause of lymphomas and other cancers.

Whilst concept has been previously briefly described as proof-of-principle, Allahyar et. al. provide a validated and detailed protocol along with accompanying analytical software (PLIER). They also perform a analytical validation using a large lymphoma sample cohort. Together, this validates a

new method for identifying translocations in non-coding regions (such as promoter translocations), and compares performance of FFPE-TLC to current FISH approaches. FFPE-TLC approach seems particularly adept at resolving complex and composite rearrangements that are otherwise difficult to interpret using conventional FISH analysis. Indeed, widespread use of this approach is likely to reveal greater complexity of these mutations in cancers than currently appreciated. The authors perform a rigorous validation, and provide detailed a detailed protocol that should encourage adoption of this useful method by clinical laboratories to diagnose this important but problematic class of mutations.

However, I believe that the manuscript could be improved by a more balanced preparation. Whilst the authors present the development and validation of FFPE-TLC as the main finding of the study, the focus of the results is instead dedicated to describing the landscape of translocations identified within the lymphoma sample cohort. As a reader, I was more interested in a detailed description and validation of the FFPE-TLC method which is often relegated to supplementary information, and less in the landscape of translocations found in the lymphoma cohort, which was foregrounded in the main text. Nevertheless, I appreciate that the authors have performed a rigorous and mature study, and my suggestions largely involve manuscript preparation, rather than additional experimental requirements.

Major comments.

1. As mentioned, my major reservation is the manuscript organization. I believe the manuscript would benefit from clear sections with headings, and a more detailed introduction and explanation of the FFPE-TLC method. There is little detail describing considerations of panel design, sequencing depth, software performance and assay reliability in the main text that would be useful to readers wishing to implement the method within their laboratory. Instead, the manuscript immediately initiates into a detailed description of translocations found within the lymphoma sample cohort. Could I suggest that the authors dedicate an initial section that walks the reader through, step by step, how a single sample is processed. From panel design, sample preparation, laboratory steps, bioinformatic analysis and final diagnosis. This initial step-by-step description would orientate the reader as to the method, before describing the subsequent validation in the lymphoma sample cohort.

I found that the description of the protocol and the software is quite brief, and later I realized that much of the detailed information I was interested in that describes the method and software, important variables and practical considerations is found within the supplementary materials. I would suggest that this relevant detail is incorporated within the main text. Given the FFPE-TLC method is the main focus of the paper, I feel this introduction would be more helpful and informative than describing the catalog of translocations found within the lymphoma sample cohort (which is of secondary importance and is largely performed to validate the method).

2. The authors perform an extensive validation of the method on a large cohort of samples. However, the analytical validation of the FFPE-TLC method would have substantially benefited from performing a set of technical replicates on a single sample. These technical replicates would enable the authors to evaluate the technical variation that occurs during the protocol and analysis, and enable the reader to understand the reliability and reproducibility of the method.

3. It would be useful if the authors could provide greater quantitative support for many of their conclusions. For example, consider line 116; "Since rearrangements with target loci juxtapose them to new flanking sequences, rearranged partner loci show an increased density of proximity-ligation sequences in FFPE-TLC and therefore can be uncovered." – what is the median increased density of proximity ligated sequences compared to negative control ? This is an illustrative example of the quantitative support throughout the manuscript that would improve the rigorosity of the studies conclusions.

4. The authors do not describe the rationale behind the design of the targeted panels for the lymphoma translocations. To help inform readers on how to design panels for use with FFPE-TLC, it

would be useful for the authors to provide in the introduction some background on the number, diversity and prevalence of noncoding translocations that are found in lymphomas and other cancers, and how panels are designed to target these sites. For example, how many translocations sites are targeted, how large are the targeted regions, and will any rare or novel translocations be missed? Does performance such as enrichment, sensitivity or specificity vary between the different targeted genes (MYC, BCL2 and BCL6) ? Does performance also vary when detecting different or novel rearrangement partners (KYNUTEX41 TBL1XR1, IKZF1 and the TOX locus) ?

5. Whilst I consider the FFPE-TLC to be an innovative method, one reservation is that it requires clinical laboratories to establish a dedicated protocol and workflow for this particular class of mutation. Accordingly, it would be useful if the authors could comment on whether the FFPE-TLC method could be incorporated within conventional targeted NGS workflows. For example, can the authors perform accurate SNP detection on the crosslinked FFPE samples or does crosslinking prevent this? Are the authors also able to identify germline/somatic mutations within their TLC-FFPE libraries? The authors note that the TLC-FFPE analyses can also be designed to include the detection of CNVs – could this be demonstrated more fully using the sample datasets? The ability to design a single panel and streamlined protocol that can accurately detect mutations, CNVs, and chromosomal rearrangements from FFPE samples would be a great advantage for implementation by clinical laboratories.

6. The authors do not indicate the sequencing or library depth is required to detect translocations at a minimum sensitivity and specificity. This analysis could be performed using the current data by subsampling the available libraries to different depths, and evaluating the sensitivity/specificity for translocation detection. This is also useful for determining the degree to which the method can be multiplexed. Similarly, by bioinformatically mixing matched normal controls and lymphoma sample libraries, the authors can demonstrate the impact of tumor purity on sensitivity and specificity. This can complement the analysis where they perform serial dilutions of the samples (see Fig.4A-B). Similarly, the authors highlight that the targeted enrichment is an important advantage of the TLC-FFPE approach. To demonstrate this advantage, it would be useful to compare their samples a non-targeted sample. This would enable the authors to directly quantify the amount of enrichment, and the relative advantages of this targeted step within the protocol. Whilst this may be onerous in sequencing libraries and costs, it may be possibly estimated from performing a simulated analysis (using tools such as Sim3c) ?

7. Many non-coding translocations occur within repetitive sequence (such as SINE elements etc.). The authors note that the TLC-FFPE is less impacted by repetitive sequences than capture NGS. Given this may be an advantage of the approach (with many structural variants occurring in repetitive regions), it would be helpful if the authors expand and quantify these advantages, and provide a more detailed quantitative analysis of the impact of repeat regions on TLC-FFPE sensitivity and specificity. This could include a comparison of enrichment, sensitivity and performance of TLC-FFPE for translocations found in repeat regions compared to non-repeat translocations?

8. The authors note that the TLC-FFPE method is largely unaffected by practical variables in the protocol. This is very useful and important information that can guide researchers whom wish to implement the technique within their clinical laboratories., and it would be useful for the authors could expand to compare the performance (such as the sensitivity and specificity) of TLC-FFPE in comparison to these practical experimental variables, such as the tissue block age, degree of DNA fragmentation and the presence of necrosis and crushing damage. Do the authors see differences in sensitivity, specificity, reliability and reproducibility in relation to these practical laboratory variables?

9. Whilst this is beyond the scope of the manuscript, out of interest, I did wonder whether authors are able to use the FFPE-TLC to resolve somatic rearrangements and determine the repertoire of immunoglobulin loci; IGH, IGK, IGL ? i.e. Can authors resolve the use of immunoglobulin

repertoires within the samples (this may be difficult or not possible due to heterogenous nature of samples)?

Minor Comments:

10. Results section could be more easily organized with headings for different sections to orientate reader?

11. Reference 43 seems to be broken: 43. (Broad institute

12. Spelling error (Hypotetical) in Figure 2A.

13. The results describing the sensitivity and specificity of FFPE-TLC in various samples could be better presented using conventional AUC curves and precision-recall graphs. For example, Figure 4A could be better presented in this way).

14. I wouldn't suggest to start a new paragraph (section) with reference to a figure (line 390)

15. In the main Figures, it would be useful to include examples of control samples for comparison. For example, Figure 2,3 and 4 would benefit from including negative examples at the MYC, BCL2 and BCL6 loci to show what a negative result looks like?

Reviewer #1 (Remarks to the Author):

This manuscript describes the feasibility of an enhanced FFPE-Targeted Locus Capture method for the detection of structural variants and in particular gene fusions in hematological disorders. Although a version of this method had already been published, this assay presents a number of important novelties, both in the experimental design (where a crucial gene enrichment step is added using capture probes) but also describes a new analysis pipeline “PLIER” that allows the detection and visualization of those events. Generally, for accurate gene fusion detection, routine capture-based methods require a comprehensive probe design that fully captures a genomic sequence where a fusion breakpoint is predicted to occur. This includes very long non-coding regions, hampering design important regions involving BCL2/6 and MYC. Therefore, detection methods such as FFPE-TLC/PLIER represent an important diagnostic tool in the clinical setting. The manuscript is well written and very detailed.

We are very pleased to read that the reviewer appreciates the importance and novelty of our work and recognizes our efforts to produce a very detailed manuscript. We also thank the reviewer for useful and constructive feedback, which was very helpful to further improve our manuscript. Please find below how we incorporated the comments and suggestions in our point-by-point answers.

I have the following comments/suggestions to the authors:

1) It would be informative to include the duration of the PLIER analysis pipeline.

Thank you for asking, we agree that such information would be informative for the future users. We now start the results section with a more detailed step-by-step overview of the protocol, in which we have also indicated the duration of various steps, including the PLIER analysis pipeline. We also added an estimated duration of the individual data processing steps of PLIER, whenever possible.

2) Line 191: “in our collection sample”: please clarify whether you’re referring to the 149-sample cohort.

To clarify this, we now state: *“in our collection of 149 samples”*

3) Line 236: did you mean “false negative”?

Apologies for the confusion. We are referring to experiments in which we mix a rearranged with a non-rearranged sample, to ask whether PLIER still identifies the ‘diluted rearrangement’. We felt it was important to confirm that PLIER did not suddenly call other (false positive) rearrangements in this mixed background. The wording ‘false positive’ is therefore correct. We changed the wording of this section to clarify the statement.

4) Lines 242-243: “Out of the 49 samples scored MYC positive by FFPE-TLC, 47 samples were also classified as such by FISH”. This statement is discordant with Table 1. According to the table, FFPE-TLC detects 47 MYC rearrangements and 2 are negative whereas 49 samples are positive by FISH.

We understand the confusion, as both statements are true: also FFPE-TLC picks up 49 MYC rearrangements, of which 47 were previously found by FISH. This can be appreciated by adding up all numbers in the white cells of this table (30+4+1+12+2: see below).

To avoid confusion we now state: “Out of the 49 samples scored MYC positive by FFPE-TLC, 47 samples were also classified as such by FISH (Table 1), while two of these MYC rearrangements were missed by FISH. They were both rearrangements in cis, with partners on the same chromosome 8..”.

MYC		FFPE-TLC				
		MYC-IGH	MYC-IGL	MYC-IGK	MYC-others	MYC negative
FISH	Positive (n=49)	30	4	1	12	2
	Negative (n=75)	0	0	0	2	73
	Inconclusive (n=1)	0	0	0	0	1
	No data (n=24)	0	0	0	0	24

5) Were the 2 MYC negative samples by FFPE-TLC tested by a different method (e.g. 4C-seq) to confirm that those weren't false negatives? No additional experiments are needed but simply include the data if available.

These two samples have unfortunately not been tested by a different method. They have been re-tested with the FFPE-TLC panel and PLIER, with identical results. We agree that an independent method would have been a better proof.

6) Based on the analysis performed in this paper, I don't believe that a general statement indicating that FFPE-TLC is superior or outperforms regular NGS capture-based sequencing is very accurate. The NGS assay that was used here wouldn't capture the fusion breakpoints due to a design limitation. Therefore, those fusions are expected to be missed. There are capture-based assays that include BCL2/6, MYC fusion design (e.g. FoundationOne Heme). If such an assay (or other without a known design limitation) fails to detect a gene fusion event that's picked up by FFPE-TLC, then you can say that your assay outperforms that specific NGS capture assay. I recommend softening the language around that in the introduction/conclusion.

We appreciate this comment. We have softened our language and no longer use words like 'superior' or 'outperform' when we compare FFPE-TLC to NGS-capture. In the abstract, we now state that: “[FFPE-TLC] shows clear advantages over standard capture-NGS methods, finding rearrangements involving repetitive sequences which they typically miss”. In the discussion, we rephrased this paragraph and now mention: “Whole genome sequencing (WGS) and regular NGS-capture methods are also used to identify SVs, find fusion partners and provide detailed information on the rearrangement breakpoint. WGS is however too expensive and computationally too demanding for a tool to diagnose rearrangements in selected target genes. Also, compared to these methods FFPE-TLC offers important advantages, particularly because it is not strictly reliant on (successful pulldown and) recognition of fusion reads. Rather, FFPE-TLC measures accumulated proximity-ligation events between chromosomal intervals flanking the breakpoint to identify a rearrangement. This, as we show, enables robust detection of rearrangements missed by regular NGS-capture methods”.

7) In the discussion, please mention Whole Genome Sequencing as a very accurate method for the detection of all SVs but discuss the limitations (cost, computational requirements, etc). We thank the reviewer for the suggestion. We now mention this method as discussed above (point 6).

8) There are few typos across the document (e.g. Figure 2B “HypotHetical”)
Thank you, we corrected this.

Reviewer #2 (Remarks to the Author):

In the study “Robust detection of translocations in lymphoma FFPE samples using Targeted Locus Capture-based sequencing”, Allahyar et. al., develop and validate FFPE-TLC - a proximity-ligation based method for targeted identification of chromosomal rearrangements in FFPE-fixed lymphoma samples. This method cleverly exploits the proximal fixing of DNA that occurs during FFPE fixation in sample preparation to identify chromosomal translocations that are a cause of lymphomas and other cancers.

Whilst the concept has been previously briefly described as proof-of-principle, Allahyar et. al. a validated and detailed protocol along with accompanying analytical software (PLIER). They also perform analytical validation using a large lymphoma sample cohort. Together, this validates a new method for identifying translocations in non-coding regions (such as promoter translocations), and compares performance of FFPE-TLC to current FISH approaches. FFPE-TLC approach seems particularly adept at resolving complex and composite rearrangements that are otherwise difficult to interpret using conventional FISH analysis. Indeed, widespread use of this approach is likely to reveal greater complexity of these mutations in cancers than currently appreciated. The authors perform a rigorous validation, and provide a detailed protocol that should encourage adoption of this useful method by clinical laboratories to diagnose this important but problematic class of mutations.

However, I believe that the manuscript could be improved by a more balanced preparation. Whilst the authors present the development and validation of FFPE-TLC as the main finding of the study, the focus of the results is instead dedicated to describing the landscape of translocations identified within the lymphoma sample cohort. As a reader, I was more interested in a detailed description and validation of the FFPE-TLC method which is often relegated to supplementary information, and less in the landscape of translocations found in the lymphoma cohort, which was foregrounded in the main text. Nevertheless, I appreciate that the authors have performed a rigorous and mature study, and my suggestions largely involve manuscript preparation, rather than additional experimental requirements.

We very much appreciate that the reviewer shows excitement about our approach and acknowledges the rigorousness and maturity of our work. We understand the request for a more in-depth description of the method in the main text, rather than in the supplemental information. We have tried to make textual modifications accordingly, hopefully to present a more balanced manuscript. Also, although not strictly asked, we hope the reviewer will be pleased to see that we did perform additional experiments and analyses, inspired by his/her questions. In particular, we now include Hi-C analysis of 6 of our lymphoma samples (sequencing depth: 257M – 540M reads) for a direct comparison between untargeted Hi-C and FFPE-TLC. As the reviewer will see, we demonstrate that FFPE-TLC is a much more cost-effective method to search for rearrangements in a selected set of target genes. Overall, we

felt that the comments helped us to improve our manuscript: we hope the reviewer agrees. Please find below our point-by-point answers.

Major comments.

1. As mentioned, my major reservation is the manuscript organization. I believe the manuscript would benefit from clear sections with headings, and a more detailed introduction and explanation of the FFPE-TLC method. There is little detail describing considerations of panel design, sequencing depth, software performance and assay reliability in the main text that would be useful to readers wishing to implement the method within their laboratory. Instead, the manuscript immediately initiates into a detailed description of translocations found within the lymphoma sample cohort.

Could I suggest that the authors dedicate an initial section that walks the reader through, step by step, how a single sample is processed. From panel design, sample preparation, laboratory steps, bioinformatic analysis and final diagnosis. This initial step-by-step description would orientate the reader as to the method, before describing the subsequent validation in the lymphoma sample cohort.

I found that the description of the protocol and the software is quite brief, and later I realized that much of the detailed information I was interested in that describes the method and software, important variables and practical considerations is found within the supplementary materials. I would suggest that this relevant detail is incorporated within the main text. Given the FFPE-TLC method is the main focus of the paper, I feel this introduction would be more helpful and informative than describing the catalog of translocations found within the lymphoma sample cohort (which is of secondary importance and is largely performed to validate the method).

We are happy to read that the reviewer did find all the necessary information in our manuscript, in the text and in the supplementary materials, as we tried hard to submit a complete and thorough manuscript. But we understand he/she wishes to read more of the details of FFPE-TLC and of PLIER in the main text.

The text of our original manuscript was aiming for a format that would be most relevant to clinical scientists and to (lymphoma) diagnostics. However, we clearly also want to reach the computational genomics and proximity-ligation communities and experts, and indeed wish to enable everybody to understand the technical and analytical intricacies. Following the recommendation of the reviewer, we therefore now start the results section with a more detailed description of the method (summarizing panel design, sample preparation, laboratory steps, bioinformatic analysis and final diagnosis). We also provide a much more detailed description of PLIER, explaining the concepts behind our automated rearrangement detection software.

We hope the reviewer finds our revised manuscript more balanced.

2. The authors perform an extensive validation of the method on a large cohort of samples. However, the analytical validation of the FFPE-TLC method would have substantially benefited from performing a set of technical replicates on a single sample. These technical replicates would enable the authors to evaluate the technical variation that occurs during the protocol and analysis, and enable the reader to understand the reliability and reproducibility of the method.

We agree with the reviewer that these are important controls. In fact, we included the data in our original manuscript but now we realize that we failed to properly discuss the results. To now better refer to these data we have added the following to the results section (validation paragraph): *“To further evaluate the robustness of our approach, we included a full technical replicate (F49 and F68), twelve technical replicate samples for library preparation, capture, sequencing and PLIER and two technical replicate samples for capture, sequencing and PLIER. In all instances, the exact same partners of MYC, BCL2 and BCL6 were scored, even with remarkably similar enrichment scores (see Suppl. Table 2, in column ‘data_type’).”*

3. It would be useful if the authors could provide greater quantitative support for many of their conclusions. For example, consider line 116; *“Since rearrangements with target loci juxtapose them to new flanking sequences, rearranged partner loci show an increased density of proximity-ligation sequences in FFPE-TLC and therefore can be uncovered.”* – what is the median increased density of proximity ligated sequences compared to negative control? This is an illustrative example of the quantitative support throughout the manuscript that would improve the rigorousness of the studies conclusions.

Figure 4a (originally Figure 4b, see below) directly illustrates how the density of proximity ligation products increases at rearrangement partners compared to a negative control. We also show the butterfly plots of all identified rearrangements (in the main and suppl. Figures), which visualize the actual amounts of ligation products at the fusion partners. Figure 4a also illustrates that quantitative differences rely on tumor load (percentage cells with the rearrangement). Other parameters that would influence the density of proximity-ligation products are sample quality, sequencing depth, genomic window considered for enrichment, repetitiveness of the rearranged region, and many others. We therefore do not think that a direct comparison of ligation-products density captured at the rearranged site across positive samples and negative controls would be very informative and revealing.

Knowing these intricacies, PLIER was designed to build the background model from the same dataset and therefore account for all these parameters to accurately interpret the density of proximity-ligation products across the genome and expresses results as significance scores. We hope the reviewer agrees PLIER provides the ultimate quantitative support for the conclusions that we draw in this work.

4. The authors do not describe the rationale behind the design of the targeted panels for the lymphoma translocations.

Thank you for asking. We have now added the following text to the discussion section, to describe the rationale behind the design panels. *“In this study, we targeted genomic intervals*

of respectively 260Kb, 330Kb and 1.05Mb around the BCL2, BCL6 and MYC genes, i.e., regions that span previously identified rearrangement breakpoints in lymphoma^{8, 28}. A tiled probe design was used, but for selective pulldown of proximity-ligated products probes may also be designed to only flank the (NlaIII) restriction enzyme recognition sites of interest³⁶. In general, for FFPE-TLC we recommend having probes at all restriction sites across the entire gene or locus of interest, plus at least 20Kb of its flanking sequences. As explained, by having sufficient proximity ligation information from flanking sequences, butterfly plots enable to unambiguously determine whether PLIER-identified chromosomal regions represent rearrangement partners fused directly to sequences inside the gene or locus of interest.”

To help inform readers on how to design panels for use with FFPE-TLC, it would be useful for the authors to provide in the introduction some background on the number, diversity and prevalence of noncoding translocations that are found in lymphomas and other cancers, and how panels are designed to target these sites. For example, how many translocation sites are targeted, how large are the targeted regions, and will any rare or novel translocations be missed?

We thank the reviewer for raising this, we are indeed excited to inform the future readers on how to design probe panels. Routine rearrangement detection in the clinic is currently also done for soft tissue sarcoma, prostate cancer and non-small cell lung carcinoma (NSCLC) and in principle, FFPE-TLC may be able to perform this task as well. We therefore now end the discussion section by stating that:

“Future prospective studies should demonstrate how FFPE-TLC performs for other cancer types, like soft tissue sarcoma, prostate cancer and non-small cell lung carcinoma (NSCLC), which are also routinely screened in diagnostic pathology for the presence of clinically relevant chromosomal rearrangements in selected target genes. Following our design rules to have probes selectively positioned at all restriction enzyme recognition sites across a gene plus 20kb of both of its flanking sequences, it should be feasible to include over 40 genes in a single probe panel, enabling simultaneous detection of their involvement in a chromosomal rearrangement.”

Does performance such as enrichment, sensitivity or specificity) vary between the different targeted genes (MYC, BCL2 and BCL6) ? Does performance also vary when detecting different or novel rearrangement partners (KYNUTEX41 TBL1XR1, IKZF1 and the TOX locus)?

This is an interesting question. To address this, we compared the enrichment scores of all rearrangement partners of BCL2, BCL6 and MYC and discussed this in the results section. We now mention: *“For all three target genes, we found somewhat higher enrichment scores for the immunoglobulin than the non-IG rearrangement partners (Suppl. Figure 9 and Suppl. Table 2), presumably because our probe design also targeted (and enriched for) the IG loci. Further, MYC rearrangements less often received extreme (>60) enrichment scores, which is probably because we probed a much larger window around MYC (>1 Mb) than around BCL2 and BCL6 (260-330 Kb): with increased distance to the breakpoint the rearrangement signal is expected to diffuse.”*. It should be reminded that in all cases discussed above, PLIER readily finds the rearrangements.

5. Whilst I consider the FFPE-TLC to be an innovative method, one reservation is that it requires clinical laboratories to establish a dedicate protocol and workflow for this particular

class of mutation. Accordingly, it would be useful if the authors could comment on whether the FFPE-TLC method could be incorporated within conventional targeted NGS workflows. For example, can the authors perform accurate SNP detection on the crosslinked FFPE samples or does crosslinking prevent this? Are the authors also able to identify germline/somatic mutations within their TLC-FFPE libraries?

We appreciate this comment, as we understand that more comprehensive genetic analysis (although still in research), is relevant for better patient specification stratification in lymphoma and for several other cancer types. We recognize the advantage of having a single protocol that can detect all types of genetic variants in a selected series of genes. Regular capture protocols already serve for the detection of SNVs and CNVs in FFPE material (but, as discussed, have difficulties finding rearrangements). As we discuss now more extensively: *“Since FFPE-TLC is based on regular capture protocols, we anticipate that FFPE-TLC analyses can also be designed to include the detection of clinically relevant SNVs and CNVs. This offers the possibility to develop methodology for the comprehensive diagnosis of all diagnostically relevant genetic variants.....//... For additional detection of clinically relevant SNVs and mutations, the recommendation would be to include tiling probes across the exons of relevant target genes.”*

We hope the reviewer finds this sufficient, as SNV and CNV detection has not been assessed within this study: we focused on the development of a robust method for rearrangement detection, which itself already was a huge effort.

The authors note that the FFPE-TLC analyses can also be designed to include the detection of CNVs – could this be demonstrated this more fully using the sample datasets? The ability to design a single panel and streamlined protocol that can accurately detect mutations, CNVs, and chromosomal rearrangements from FFPE samples would be a great advantage for implementation by clinical laboratories.

This is related to the previous comment, so please see our answer above.

6. The authors do not indicate the sequencing or library depth is required to detect translocations at a minimum sensitivity and specificity. This analysis could be performed using the current data by subsampling the available libraries to different depths, and evaluating the sensitivity/specificity for translocation detection. This is also useful for determining the degree to which the method can be multiplexed. Similarly, by bioinformatically mixing matched normal controls and lymphoma sample libraries, the authors can demonstrate the impact of tumor purity on sensitivity and specificity. This can complement the analysis where they perform serial dilutions of the samples (see Fig.4A-B).

Regarding assessing the impact of tumor purity on sensitivity and specificity, we indeed wish to refer to figure 4: here we systematically analyze in mixed populations of cells the minimal percentage of cells that are required to have a successful detection of the rearrangement by PLIER (with 5% rearranged cells, the rearrangement is robustly detected). Specificity is unaltered: we find no false positive calls, as described. To address the question of required sequencing depth, we have now in silico down sampled these datasets to determine the minimum sequencing depth required for successful detection of the rearrangement by PLIER. The result of this comprehensive analysis is now added as a new panel in Figure 4 (as Figure 4c). We now discuss in the results section: *“As shown in Figure 4C, in the undiluted tumor samples not more than 75K on-target reads were needed to robustly detect the MYC, BCL2*

and BCL6 rearrangements. When present in only 5% of the cells, one million on-target reads were sufficient for their detection."

Similarly, the authors highlight that the targeted enrichment is an important advantage of the FFPE-TLC approach. To demonstrate this advantage, it would be useful to compare their samples to a non-targeted sample. This would enable the authors to directly quantify the amount of enrichment, and the relative advantages of this targeted step within the protocol. Whilst this may be onerous in sequencing libraries and costs, it may be possibly estimated from performing a simulated analysis (using tools such as Sim3c)?

Although this conceptually is of course the obvious advantage that justifies all capture-based sequencing applications, we are happy to provide the reviewer with this comparison. We have now performed Hi-C on six of our lymphoma samples and describe the results:

"For further validation and to explore alternative proximity-ligation methods, we processed six lymphoma samples by Hi-C. Despite much deeper sequencing (257M-540M Hi-C read pairs, compared to 17M-71M read pairs sequenced for FFPE-TLC), Hi-C failed to detect the known rearrangements and the number of captured ligation-products at the rearrangement site was very limited (Suppl. Figure 8)".

7. Many non-coding translocations occur within repetitive sequences (such as SINE elements etc.). The authors note that the TLC-FFPE is less impacted by repetitive sequences than capture NGS. Given this may be an advantage of the approach (with many structural variants occurring in repetitive regions), it would be helpful if the authors expand and quantify these advantages, and provide a more detailed quantitative analysis of the impact of repeat regions on TLC-FFPE sensitivity and specificity. This could include a comparison of enrichment, sensitivity and performance of TLC-FFPE for translocations found in repeat regions compared to non-repeat translocations?

We have three examples in our collection where capture-NGS failed to find the translocation because of a break in repetitive sequences. For these, we checked how robustly they were detected by FFPE-TLC. We now describe in the results section: *"In three other cases (F191, F192, F198) Capture-NGS was not able to identify the rearrangement partner as the breakpoint has occurred at a non-unique sequence, whereas FFPE-TLC readily scored them (z-scores >60)".* We further refer to Figure 5D, in which we analyze what percentage of the total collection of breakpoints identified in our studies would likely be missed by capture-NGS due to repetitiveness.

8. The authors note that the TLC-FFPE method is largely unaffected by practical variables in the protocol. This is very useful and important information that can guide researchers who wish to implement the technique within their clinical laboratories., and it would be useful for the authors could expand to compare the performance (such as the sensitivity and specificity) of TLC-FFPE in comparison to these practical experimental variables, such as the tissue block age, degree of DNA fragmentation and the presence of necrosis and crushing damage. Do the authors see differences in sensitivity, specificity, reliability and reproducibility in relation to these practical laboratory variables?

Unfortunately, we do not have access to comprehensive quantitative data regarding the degree of DNA fragmentation and the presence of necrosis and crushing damage. However, we have 5 participating centers where each has their own protocol for FFPE processing and storage. This variation may well impact the sample quality differently. Yet, when we stratify

samples on their center of origin, even though we observe this difference, we find no differences in the sensitivity and specificity of FFPE-TLC (PLIER) and all calls are well above the significance threshold of 8.0. The same is true for tissue block age. We now mention and show this in the discussion section, as follows: *“No major differences in sensitivity and specificity were found between the FFPE samples provided by the five different clinical centers, showing that FFPE-TLC is resistant to the differences that may exist between their protocols for FFPE preparation and storage. Also, FFPE-TLC performed similarly on recent and older tissue blocks (Suppl. Figure 10).”*

9. Whilst this is beyond the scope of the manuscript, out of interest, I did wonder whether authors are able to use the FFPE-TLC to resolve somatic rearrangements and determine the repertoire of immunoglobulin loci; IGH, IGK, IGL? i.e. Can authors resolve the use of immunoglobulin repertoires within the samples (this may be difficult or not possible due to the heterogeneous nature of samples)?

This is an interesting question. The IG patterns that we see in the butterfly plots must be clonal, representing the identity of the IG allele at the stage when it recombined with one of our target genes (MYC, BCL2 or BCL6). Clonal selection of this (cancer) cell allows its accurate analysis. However, assessing the repertoire of IG alleles is not possible (indeed because of heterogeneity).

Minor Comments:

10. Results section could be more easily organized with headings for different sections to orientate the reader?

Subheadings were added. Thank you for pointing this out.

11. Reference 43 seems to be broken: 43. (Broad institute
Fixed

12. Spelling error (Hypotetical) in Figure 2A.

Done

13. The results describing the sensitivity and specificity of FFPE-TLC in various samples could be better presented using conventional AUC curves and precision-recall graphs. For example, Figure 4A could be better presented in this way).

Thank you for pointing this possibility out. Indeed, we used precision-recall curves when we were investigating the optimal significance threshold of PLIER as well as during the hyper-parameter optimization procedure (see **Parameter optimization for PLIER**). We updated the text to better clarify this.

It is worth noting that our analysis is not a standard machine learning training procedure. Specifically in our work, a single true negative call occurs when the PLIER finds no rearrangement in a given control sample, but in fact all bins in that control sample that are not found significant could be considered as true negative calls. This consideration is justified because if those bins would have been found significant by PLIER, we would have considered all of those bins to be false positive calls. In our setup, during the precision-recall curve estimation, when reducing the significance threshold, PLIER may call many false-positives (essentially every bin in the genome if significance threshold becomes very small). Such false-positive calls were not designated to be true-negatives when the threshold was higher. This

situation does not occur in a standard machine learning training procedure where a single new positive call by the “classifier” (after reducing the threshold) is either false-positive or true positive. In our case however, the called bin is either true positive, or undefined. Regardless of the above difference, we now include the precision-recall curve as **Suppl. Figure 11**. You may also see this figure below. The numbers within each circle indicates the threshold used to identify the call types (e.g., true positive or false-positives). As described in the manuscript, we chose a significance threshold of 8.0 to stay below 1% FDR.

14. I wouldn't suggest to start a new paragraph (section) with reference to a figure (line 390)
Fixed

15. In the main Figures, it would be useful to include examples of control samples for comparison. For example, Figure 2,3 and 4 would benefit from including negative examples at the MYC, BCL2 and BCL6 loci to show what a negative result looks like?

It should be noted that visualizing a negative example would give a rather empty butterfly plot. For comparison, the reader can refer to the “empty” quadrants of the already shown butterfly plots. We therefore believe that visualizing negative samples in a butterfly fashion would not be too informative. We would be happy to include such examples as an additional supplementary figure if the reviewer feels that such visualizations are required.

Reviewers' Comments:

Reviewer #1:

Remarks to the Author:

I would like to thank the authors for meticulously addressing my comments and providing all the necessary clarifications. I don't have any more questions.

Reviewer #2:

Remarks to the Author:

I agree that the authors have demonstrated the concept, implementation, performance and advantages of the FFPE-TLC/PLIER method for detecting translocations. I also appreciate the effort the authors have undertaken to respond to my comments. Accordingly, I would like to congratulate the authors on a great paper and great technology, and endorse publication of their manuscript.